# Electrochemical $CO_2$ reduction to ethylene by ultrathin CuO nanoplate arrays

Wei Liu[1], Pengbo Zhai[2], Aowen Li[3,4], Bo Wei[1], Kunpeng Si[1], Yi Wei[5,6], Xingguo Wang[1], Guangda Zhu[7], Qian Chen[1], Xiaokang Gu[1], Ruifeng Zhang[1], Wu Zhou [3,4,8] & Yongji Gong [1,9 ✉]

Electrochemical reduction of $CO_2$ to multi-carbon fuels and chemical feedstocks is an appealing approach to mitigate excessive $CO_2$ emissions. However, the reported catalysts always show either a low Faradaic efficiency of the $C_{2+}$ product or poor long-term stability. Herein, we report a facile and scalable anodic corrosion method to synthesize oxygen-rich ultrathin CuO nanoplate arrays, which form $Cu/Cu_2O$ heterogeneous interfaces through self-evolution during electrocatalysis. The catalyst exhibits a high $C_2H_4$ Faradaic efficiency of 84.5%, stable electrolysis for ~55 h in a flow cell using a neutral KCl electrolyte, and a full-cell ethylene energy efficiency of 27.6% at 200 mA cm$^{-2}$ in a membrane electrode assembly electrolyzer. Mechanism analyses reveal that the stable nanostructures, stable $Cu/Cu_2O$ interfaces, and enhanced adsorption of the *OCCOH intermediate preserve selective and prolonged $C_2H_4$ production. The robust and scalable produced catalyst coupled with mild electrolytic conditions facilitates the practical application of electrochemical $CO_2$ reduction.

[1] School of Materials Science and Engineering, Beihang University, Beijing 100191, China. [2] College of Physics, Qingdao University, Qingdao 266071, China. [3] School of Physical Sciences, University of Chinese Academy of Sciences, Beijing 100049, China. [4] CAS Key Laboratory of Vacuum Physics, University of Chinese Academy of Sciences, Beijing 100049, China. [5] State Key Laboratory of Organic-Inorganic Composites, Beijing University of Chemical Technology, Beijing 100029, China. [6] Beijing Key Laboratory of Electrochemical Process and Technology for Materials, Beijing University of Chemical Technology, Beijing 100029, China. [7] Beijing National Laboratory for Molecular Sciences, Laboratory of Polymer Physics and Chemistry, Institute of Chemistry, Chinese Academy of Sciences, Beijing 100190, China. [8] CAS Center for Excellence in Topological Quantum Computation, University of Chinese Academy of Sciences, Beijing 100049, China. [9] Center for Micro-Nano Innovation of Beihang University, Beijing 100191, China. ✉email: yongjigong@buaa.edu.cn

The depletion of fossil fuels and their non-renewability continue to emphasize the importance of utilizing renewable energy to convert carbon dioxide ($CO_2$) into fuels and chemical feedstocks[1–3]. Artificial conversion of $CO_2$ is essential to reduce its emissions and realize the sustainable development of humanity[3,4]. The electrochemical $CO_2$ reduction reaction ($CO_2RR$) is the most attractive method due to its mild reaction conditions and capacity for renewable electricity storage[5]. Among the products of the $CO_2RR$, low-value $C_1$ species such as carbon monoxide (CO) and formate ($HCOO^-$) are the most common products due to the sluggish kinetics of the C–C coupling reaction[6,7]. As shown by previous studies, the C–C coupling reaction is the vital step for $C_{2+}$ species formation[8–11]. Theoretical calculations reveal that excessively strong or weak binding of *CO intermediates is unfavorable for the generation of $C_{2+}$ species[9,12–14]. Copper-based (Cu-based) materials possessing a moderate adsorption energy of the *CO intermediate have been the most efficient catalysts in converting $CO_2$ to $C_{2+}$ hydrocarbons and oxygenates with considerable activity[2,15–18]. The $CO_2RR$ to ethylene ($C_2H_4$) with high current density and Faradic efficiency (FE) is intensively studied because of the extremely high industrial value and limited sources of $C_2H_4$[18–21]. However, as multi-step electron and proton transfer processes are involved in $C_2H_4$ formation, hydrogen ($H_2$) and other by-products such as methane ($CH_4$) will inevitably be produced during electrolysis[16,20,22–24]. Therefore, the activity and selectivity of $C_2H_4$ formation are still limited for Cu-based catalysts.

Various procedures such as facet and grain boundary regulation[25–27], morphology engineering[18,28–32], electrode surface additive modification[14,33–36], electrolyte design[37,38] and oxide-derived catalysis[16,20,39–42] have been proposed to improve the current density and FE of $C_2H_4$ production. Among these catalysts, oxide-derived Cu (OD-Cu) has the highest product ratio of $C_2H_4$ to $CH_4$[21,26,40]. These catalysts have generally been synthesized by thermal oxidation and subsequently reduced through $H_2$ annealing or in situ electrolysis. Hemma et al. revealed that the increased local pH caused by the rough morphology of OD-Cu suppressed $CH_4$ formation[21]. The existence of $Cu^+$ species may promote the adsorption of $CO_2RR$ intermediates, which is crucial to improve the $C_2H_4$ selectivity[21]. Although the existence and important role of $Cu^+$ species in the $CO_2RR$ have been proven by many in situ and *operando* tests (Raman spectroscopy, X-ray absorption spectroscopy, etc.)[20,26,43,44], thermal oxidation treatments usually result in a disordered morphology and an inadequate amount of oxidation species[40]. These issues lead to the disappearance of $Cu/Cu^{\delta+}$ heterointerfaces during long-term electrocatalysis tests due to the self-evolution of surface species and morphologies. Therefore, constructing stable $Cu/Cu^{\delta+}$ heterointerfaces with exquisite nanostructures is highly desirable to improve the stability of $C_2H_4$ production.

Here, we report the preparation of an OD-Cu catalyst with a dense vertical lamellate Cu nanostructure (denoted as DVL-Cu). The DVL-Cu catalyst was obtained via in situ electrochemical reduction of CuO ultrathin nanoplate arrays (denoted as CuO-NPs) synthesized by galvanostatic anodic oxidation in an alkaline solution. The electrochemical $CO_2RR$ test of DVL-Cu delivered an $C_2H_4$ FE of 73.6% and a total $C_{2+}$ (mainly ethylene and ethanol) FE of >80% at −0.8 V vs. the reversible hydrogen electrode (RHE, all potentials are with respect to this reference in this article) in an H-cell and an even higher $C_2H_4$ FE of 84.5% in a flow cell, with neutral potassium chloride (KCl) as the catholyte. Moreover, the catalyst achieved an ethylene energy efficiency ($EE_{C_2H_4}$) of 28.9% and ~55 h stable long-term electrolysis in a flow cell. The experimental and simulation results reveal that the nanostructured DVL-Cu generated $Cu/Cu_2O$ heterogeneous interfaces and dispersed the electrode current density effectively to avoid agglomeration during the $CO_2RR$. Meanwhile, the KCl electrolyte impedes the dissolution/redeposition of $Cu^+$ species due to its high local pH microenvironment and suppresses hydrogen evolution at higher overpotentials, which favors the high selectivity and stability of the DVL-Cu catalyst. Density functional theory (DFT) calculations suggest that the $Cu/Cu_2O$ interfaces in DVL-Cu facilitate $C_2H_4$ formation due to their reduced C–C dimerization energy in the $C_2H_4$ formation pathway. The facile synthetic method, mild electrolysis conditions (neutral electrolyte), and prolonged electrolysis stability make this material a promising candidate for commercial $CO_2RR$ catalysts in the future.

## Results

**Preparation and characterization of the DVL-Cu catalyst.** First, CuO-NPs were synthesized by galvanostatic anodic oxidation of Cu foil in a 1 M sodium hydroxide (NaOH) electrolyte. Copper hydroxide ($Cu(OH)_2$) was primarily generated on the Cu surface during anodic oxidation[45], while $Cu(OH)_2$ spontaneously converted to CuO at a proper current density. We found that 0.26 mA cm$^{-2}$ (calculated based on the geometric area) was the most suitable current density for CuO-NPs synthesis, resulting in the best catalyst performance. Excessive current with rapid $Cu(OH)_2$ production led to the complete coverage of blue $Cu(OH)_2$ on the surface, and an insufficient current generated only a thin oxide layer rather than nanoplate arrays.

A schematic illustration of the CuO-NPs formation process is displayed in Fig. 1a. This process can be divided into three stages: the Cu etching stage, CuO nucleation stage and CuO growth stage. The Cu surface was first corroded by a positive bias at the etching stage (Fig. 1b). Once $Cu^{2+}$ was saturated in the solution, CuO nucleated and grew on the surface as $Cu(OH)_2$ was generated and dehydrated (Supplementary Fig. 1). The surface oxygen content no longer increased after the nucleation stage, indicating that the surface had been completely covered by the oxide layer (Fig. 1c). Eventually, vertically arranged and densely stacked CuO-NPs were synthesized. The successful synthesis of CuO-NPs on a large piece of Cu foil ($25 \times 25$ cm$^2$) by this method proves the scalability to prepare catalysts for industrial applications (Supplementary Figs. 2–4, the samples used for characterization were obtained from Cu foils with a size of $1.5 \times 3$ cm$^2$ unless otherwise specified). The morphology and atomic structure of an individual CuO nanoplate before and after the $CO_2RR$ can be characterized in detail after sonication. As shown by transmission electron microscopy (TEM) images, the CuO nanoplates are composed of polygonal ultrathin CuO nanosheets (Fig. 1d and Supplementary Fig. 5). The nm-scale thickness of the CuO nanosheets was confirmed by atomic force microscopy (AFM), showing that the thicknesses of the corresponding layers are 3.136 and 0.782 nm, respectively (Fig. 1e). The ultrathin thickness of CuO nanosheets leads to ultrafine Cu nanostructures with numerous exposed catalytically active sites during electrocatalysis. Specifically, the thickness of the whole CuO-NPs layer on Cu foil is 2.25 μm, as measured by the scanning electron microscopy (SEM) cross-sectional image (Supplementary Fig. 6), which also exhibits a uniform and compact stacking pattern. Furthermore, this three-dimensional self-supporting structure is beneficial to mass transfer and the exposure of catalytically active sites. In addition, the CuO-NPs were thermally reduced under Ar/$H_2$ to obtain reduced-CuO-NPs (denoted as R-CuO-NPs, Supplementary Fig. 7) as a control group to test the $CO_2RR$ performance.

The DVL-Cu catalyst was produced during the in situ electrochemical $CO_2RR$ by self-evolution in a gas-tight H-cell (Fig. 2a). CuO-NPs were used as the working electrode, a

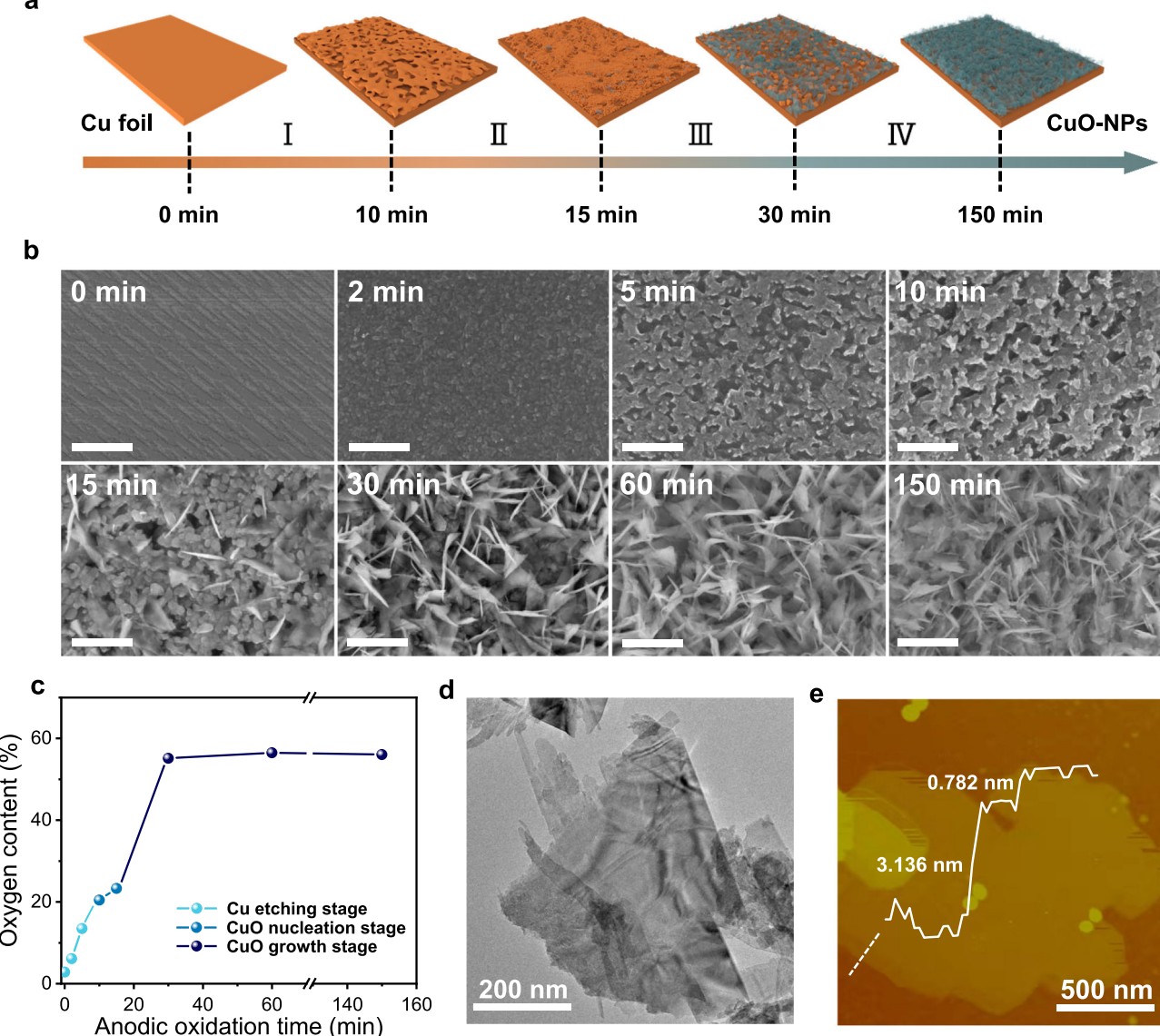

**Fig. 1 Preparation and characterization of CuO-NPs. a** Schematic illustration of the preparation of CuO-NPs. **b** SEM images of Cu foils at different anodic oxidation times. The scale bars are 500 nm. **c** Oxygen content of the Cu surface at different anodic oxidation times. **d** TEM image of the CuO-NPs. **e** AFM image of the CuO-NPs and the corresponding height profile from the dashed line.

Ag/AgCl electrode was used as the reference electrode (RE), and platinum foil was employed as the counter electrode. In situ galvanostatic reduction was performed at $-0.8$ V under a $CO_2$ atmosphere. The black foil quickly turned rufous after 20 s of electrolysis (Supplementary Fig. 8), suggesting that the CuO-NPs were reduced to a lower valence. After 20 min of electrolysis, the vertically arranged and densely stacked laminated nanostructure was retained (Fig. 2b, c, and Supplementary Figs. 9, 10). However, every single piece of ultrathin CuO nanoplate was converted to a thicker nanoplate composed of small nanoparticles. These nanoparticles were considered active catalytic sites for $C_2H_4$ formation, as suggested by a previous study[46]. This nanoplate-like morphology suppresses the agglomeration of Cu nanoparticles, which benefits the long-term catalytic stability. The curves depicted in Fig. 2d present the grazing incident X-ray diffraction (GI-XRD) results of the original CuO-NPs and DVL-Cu after different reduction times. The GI-XRD result of CuO-NPs shows the successful synthesis of CuO without any other impurities (such as $Cu(OH)_2$ and $Cu_2O$). The peaks ascribed to cuprite-type

$Cu_2O$ begin to appear after 20 s of electrochemical reduction. The CuO peaks completely disappear after 1 min of electrochemical reduction, while the peak of $Cu_2O$ still remains even when the reduction time is extended to 2 h, which is consistent with previous studies on OD-Cu catalysts[20,26,45,46]. Comparing the line for the 2 h reduction time and that for the 5 min reduction time, the peak intensity of the Cu(110) facet increases with the duration of electrolysis. As Cu(111) is the most thermodynamically stable facet in polycrystalline Cu[26], the increased ratio of Cu(110) might result from the stabilizing effect of $CO_2$RR intermediates[26].

Supplementary Figs. 11, 12 and Fig. 2e show X-ray photoelectron spectroscopy (XPS) and Auger electron spectroscopy (AES) results of CuO-NPs and DVL-Cu (taken after 1 h of electrolysis) for different $Ar^+$ beam etching times, noting that the $Ar^+$ beam could etch the surface of the sample to reveal subsurface information. The typical satellite peak and Cu $LMM$ Auger peak location (971.3 eV) of the CuO-NPs indicate the characteristics of $Cu^{2+}$ species[20]. No satellite peak was found in the DVL-Cu XPS

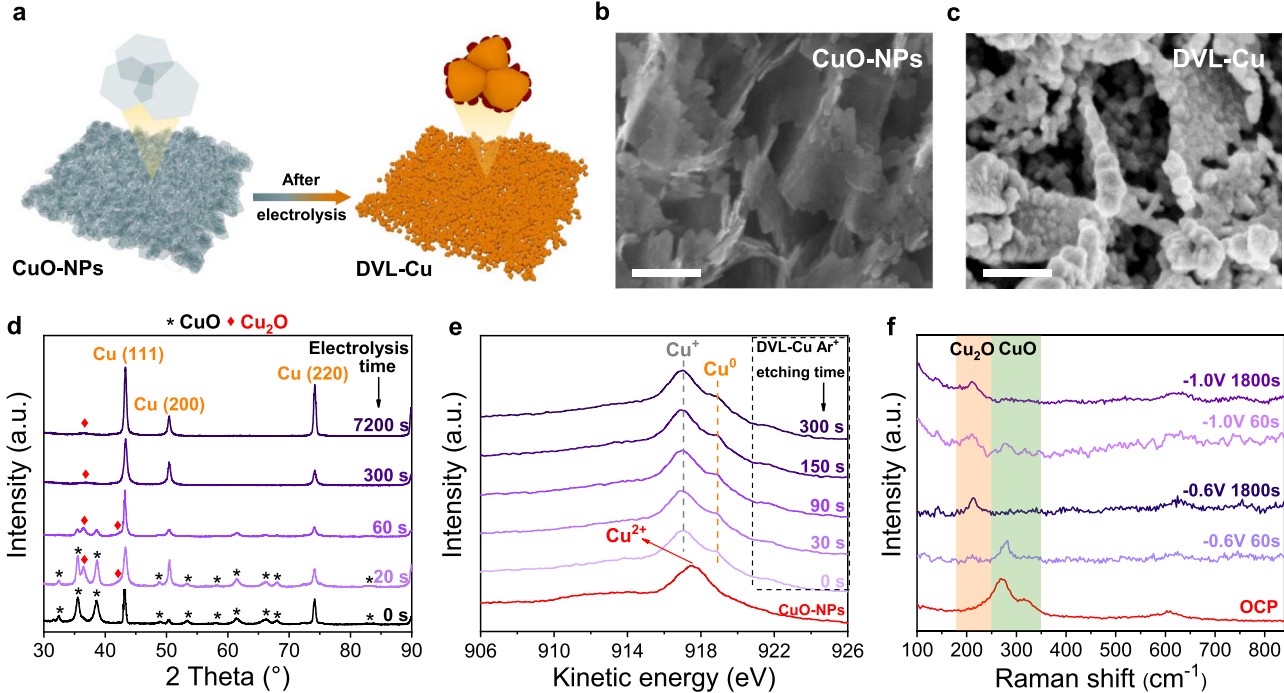

**Fig. 2 Preparation and characterization of DVL-Cu. a** Schematic illustration of morphology changes during the electrochemical reduction of DVL-Cu and CuO-NPs. **b, c** High-magnification SEM images of CuO-NPs (**b**) and DVL-Cu (**c**). The scale bars are 200 nm. **d** GI-XRD patterns of CuO-NPs and DVL-Cu for different reduction times. **e** Cu *LMM* Auger spectra of CuO-NPs and DVL-Cu (after 1 h reduction) with respect to different $Ar^+$ etching times. **f** In situ Raman spectra of DVL-Cu during electrolysis.

results for different etching times, suggesting that the original $Cu^{2+}$ species in the CuO-NPs are completely reduced after electrolysis. Cu Auger *LMM* spectra demonstrate that $Cu^0$ and $Cu^+$ species co-exist in DVL-Cu even after 300 s of $Ar^+$ beam etching (the etching depth was estimated to be 50 nm), which could entirely remove the surface oxide layer formed by air oxidation (usually < 5 nm). In situ Raman spectroscopy was further performed to identify the valence of Cu during electrolysis (Fig. 2f and Supplementary Fig. 13). At the open-circuit potential (OCP), peaks located at 286, 327 and 618 $cm^{-1}$ were ascribed to CuO. CuO characteristic peaks weakened, and peaks associated with $Cu_2O$ began to appear when DVL-Cu was electrolyzed at −0.6 V for 60 s. When the electrolysis was carried out at −0.6 V for 1800 s, the peaks of $Cu_2O$ still existed, but those of CuO completely disappeared. These results were similar for electrolysis at a higher overpotential of −1.0 V, except that the peak intensity of $Cu_2O$ was lower than that at −0.6 V. Thus, it was concluded that $Cu^+$ species were preserved during the electrochemical $CO_2RR$, which might be due to stabilization effects generated by the high local pH and $Cl^-$ [16,21].

Additionally, aberration-corrected scanning transmission electron microscopy (STEM) combined with electron energy-loss spectrometry (EELS) was performed to reveal the valence states and distribution of Cu species in DVL-Cu (Fig. 3 and Supplementary Fig. 14). The structure of a large particle decorated by several small particles is displayed in the bright-field (BF) image (Fig. 3a). The fast Fourier transform (FFT) patterns of the central large particle (#1) and a surrounding small particle (#2) are consistent with the Cu[110] and $Cu_2O$[110] zone axes, respectively. Cu(1$\bar{1}$1) and (002) facets were delineated in the enlarged image of area #1 (Fig. 3b). Energy-loss near-edge fine structure (ELNES) analysis, which is a commonly applied method to determine the valence state of 3d metal elements, was also performed on the DVL-Cu sample to confirm its composition and distribution[47]. The fine structures of the EELS spectrum

extracted from area #1 are consistent with $Cu^0$, while the spectrum from area #2 is related to $Cu^+$, matching well with the FFT analysis (Fig. 3c). Cu valance state mapping based on EELS spectrum imaging shows the distribution of Cu and $Cu_2O$ in the whole area (Fig. 3d). The dashed lines highlighted in the overlay of the Cu and $Cu_2O$ maps indicate the existence of $Cu/Cu_2O$ interfaces on DVL-Cu, which might be beneficial for the $CO_2RR$.

**Electrochemical $CO_2RR$ performance in the H-cell.** To measure the electrochemical $CO_2RR$ performance of DVL-Cu, we performed electrolysis in $CO_2$-saturated 0.5 M KCl by using a gas-tight H-cell. For comparison, the electrochemical $CO_2RR$ performance of the R-CuO-NPs was also evaluated under the same conditions. It is worth noting that KCl was chosen as the electrolyte because it benefits the preservation of $Cu_2O$ during the $CO_2RR$ through a high local pH, and the specific adsorption of $Cl^-$ suppresses hydrogen evolution at higher overpotentials[48,49].

The geometric current density of DVL-Cu in the $CO_2$-saturated electrolyte is considerably higher than that of R-CuO-NPs over the whole test potential window (Supplementary Fig. 15), while the two catalysts deliver similar geometric current densities in the Ar-saturated electrolyte, indicating that DVL-Cu possesses higher intrinsic $CO_2RR$ activity. To exclude the electrochemical surface area (ECSA) influence, we calculated the roughness factor (RF) of the two samples by the double-layer capacitance method (Supplementary Fig. 17). DVL-Cu has an RF of 201.7, while R-CuO-NPs have a comparable RF of 165.7. These measurements eliminate the influence of the ECSA and confirm the higher intrinsic $CO_2RR$ activity of DVL-Cu than R-CuO-NPs. Moreover, DVL-Cu exhibits high $C_2H_4$ and $FE_{C_{2+}}$ FE, which shows a higher $FE_{C_2H_4}$ of 74.9 ± 2.6% and $FE_{C_{2+}}$ of 80.5 ± 2.3% at −0.9 V (Fig. 4a and Supplementary Table 3) than that of R-CuO-NPs ($FE_{C_2H_4}$ = 52.0 ± 3.7%, $FE_{C_{2+}}$ = 61.9 ± 4.0%) (Fig. 4b). The liquid products of DVL-Cu and R-CuO-NPs are mainly composed

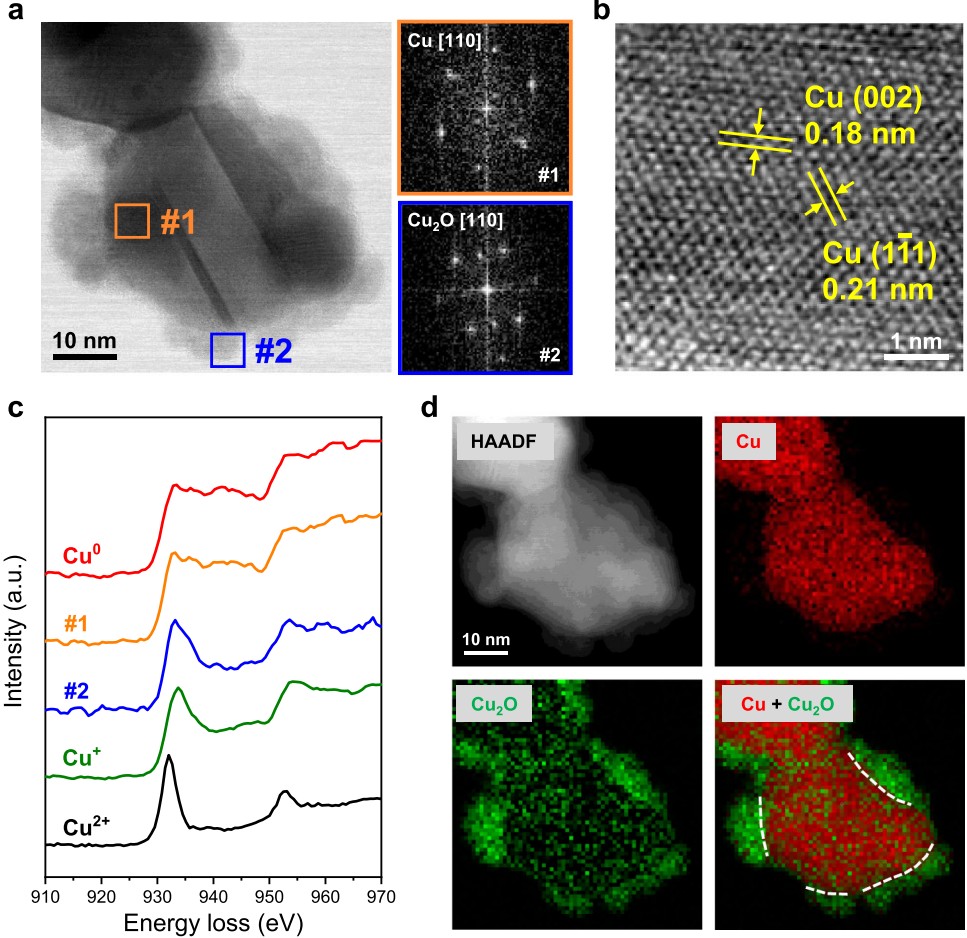

**Fig. 3 STEM and EELS characterizations of the DVL-Cu catalyst. a** STEM BF image (left) of DVL-Cu and FFT patterns (right) of the corresponding area in the BF image. **b** Enlarged image of area #1 in (**a**). **c** EELS spectra acquired from areas #1 and #2 in (**a**). Standard Cu, Cu$^+$ and Cu$^{2+}$ EELS results are plotted as references. **d** STEM high-angle annular dark-field (HAADF) image and EELS maps of Cu and Cu$_2$O and their overlay in DVL-Cu. The dashed lines highlighted in the overlay denote the Cu/Cu$_2$O interfaces.

of ethanol and formate, with a small percentage of acetate, propanol, etc. (Fig. 4e). Furthermore, the $FE_{H_2}$ is only 13.6% at −0.9 V of DVL-Cu, which might be due to the massive consumption of H atoms in the proton-coupled electron transfer (PCET) step during the CO$_2$RR. Additionally, the partial current density curves and C$_2$H$_4$ Tafel curves were plotted to reveal the kinetic characteristics. The C$_1$ partial current densities of these two catalysts are comparable, but the C$_{2+}$ partial current density of DVL-Cu is five times higher than that of R-CuO-NPs at −1.0 V (Fig. 4c). The C$_2$H$_4$ Tafel slopes of DVL-Cu and R-CuO-NPs are 179.3 mV decade$^{-1}$ and 421.1 mV decade$^{-1}$ (Fig. 4d), respectively, suggesting a much lower C$_2$H$_4$ formation kinetic barrier on DVL-Cu than on R-CuO-NPs. The sluggish kinetics of C$_2$H$_4$ and C$_{2+}$ formation on R-CuO-NPs was further confirmed by the C$_{2+}$/ C$_1$ (CO, formate) and C$_2$H$_4$/CH$_4$ ratios in CO$_2$RR products (Fig. 4f). The C$_{2+}$/C$_1$ and C$_2$H$_4$/CH$_4$ product ratios of DVL-Cu at −0.8 V are 19.4 and 208.6, respectively, indicating that most of the CO intermediates tend to dimerize, generating C$_{2+}$ products rather than forming C$_1$ species. In comparison, the C$_{2+}$/C$_1$ and C$_2$H$_4$/CH$_4$ product ratios (3.6 and 7.0 at −0.8 V) on R-CuO-NPs are much lower than that of DVL-Cu, which verifies the sluggish kinetics of the C–C coupling step on R-CuO-NPs.

The performance observed in stability tests with OD-Cu catalysts was always unsatisfactory according to previous studies. We performed long-term electrolysis under potentiostatic mode with the same conditions mentioned above (Supplementary

Figs. 16 and 18). DVL-Cu shows a steady $i$-t curve and high C$_2$H$_4$ selectivity ($FE_{C_2H_4}$ = 62.2%) for 50 h at −0.8 V. In contrast, R-CuO-NPs only retains a stable $FE_{C_2H_4}$ for less than 7 h at −0.8 V. Post-electrolysis SEM images of DVL-Cu show that nanoplates densely stack in order without any agglomeration after 50 h of electrolysis, and only a slight increase in plate thickness is observed (Supplementary Figs. 25a and b). The poor stability of R-CuO-NPs might be caused by the agglomeration of Cu nanoparticles during the CO$_2$RR (Supplementary Fig. 19).

Additionally, the high-resolution transmission electron microscopy (HRTEM) images confirm that there are no Cu$_2$O species in R-CuO-NPs after 2 h of the CO$_2$RR, either at the edge or the center (Supplementary Fig. 20), and the exposed plane is similar to that of DVL-Cu. Cu$_2$O characteristic peaks are also absent in the GI-XRD profiles of R-CuO-NPs after 1 h of the CO$_2$RR (Supplementary Fig. 21). However, the DVL-Cu sample preserves a similar Cu/Cu$_2$O composite structure and subsurface Cu$^+$ content after a long-term test (Supplementary Figs. 27–29). The presence of Cu$_2$O in DVL-Cu may be the origin of the CO$_2$RR performance under the premise that the ECSA, exposed Cu facets, and hydrophilicity (Supplementary Fig. 30) of these two catalysts are comparable.

**Electrochemical CO$_2$RR performance in the flow cell and MEA electrolyzer.** To evaluate the application potential of CuO-NPs for the industrial CO$_2$RR, we further measured their catalytic performance in a flow cell. A Cu membrane (500 nm) was

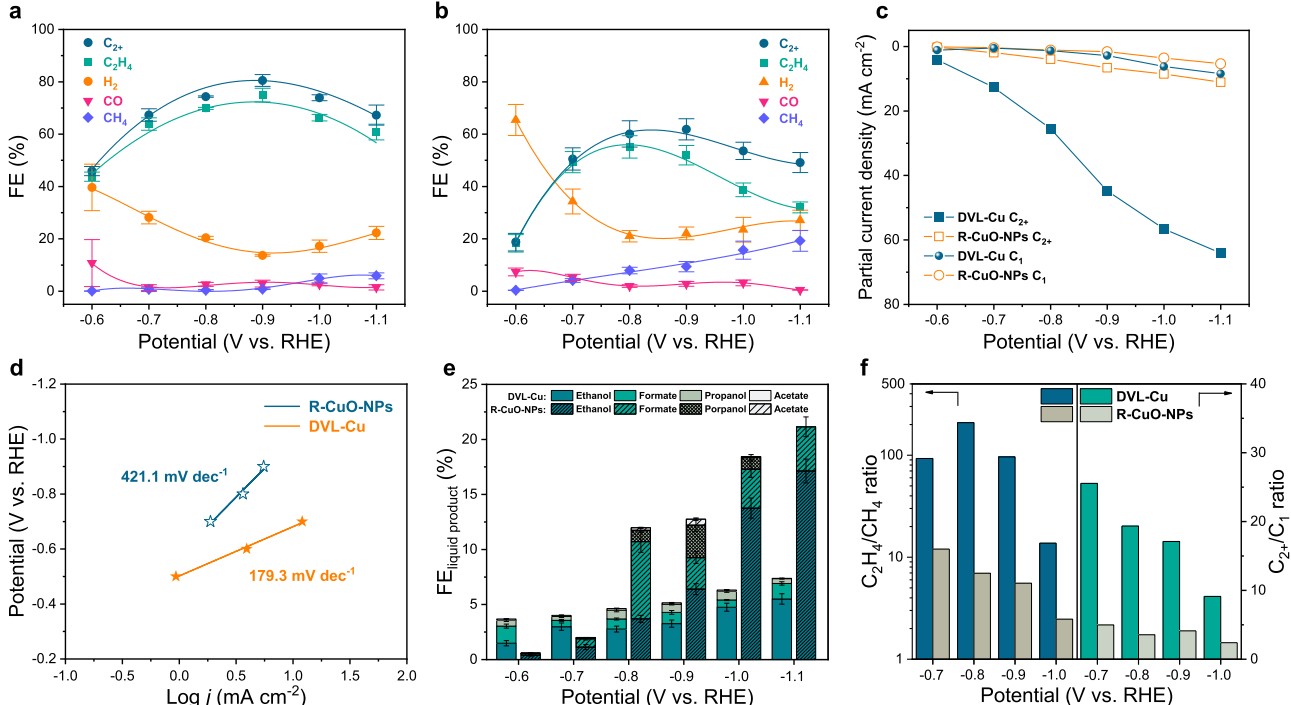

**Fig. 4 Electrochemical CO₂RR performance of DVL-Cu and R-CuO-NPs. a** FEs of DVL-Cu. **b** FEs of R-CuO-NPs. **c** $C_{2+}$ and $C_1$ partial current densities of DVL-Cu and R-CuO-NPs. **d** $C_2H_4$ partial current density Tafel plots of DVL-Cu and R-CuO-NPs. **e** FEs of liquid products of DVL-Cu and R-CuO-NPs. **f** $C_2H_4/CH_4$ ratios and $C_{2+}/C_1$ ratios for CO₂RR products of DVL-Cu and R-CuO-NPs. Error bars represent the standard deviation of three independent measurements.

deposited on carbon paper (Sigracet 28BC) with a gas diffusion layer (GDL) by electron beam evaporation (Fig. 5a). Then, galvanostatic anodic oxidation was performed to obtain CuO-NPs on the GDL (CuO-NPs@GDL), similar to those on copper foil. The synthesis and characterization details can be seen in the supplementary information (Methods and Supplementary Figs. 32 and 33). The as-fabricated electrode was utilized in a flow cell setup reported previously (Supplementary Figs. 34 and 35), where KCl and KOH served as the catholyte and anolyte, respectively. CuO nanoplates on the GDL were converted to DVL-Cu during the CO₂RR with a maintained plate-like nanostructure (DVL-Cu@GDL, Supplementary Fig. 36).

The CO₂RR performance and cathodic $EE_{C_2H_4}$ are displayed in Fig. 5b. For $C_{2+}$ species, the FEs increase from $62.3 \pm 1.4\%$ (at −0.63 V) to $85.4 \pm 2.0\%$ (at −0.81 V) and then drop to $69.3 \pm 3.2\%$ (at −1.01 V) due to the increased hydrogen generation at a large overpotential. Remarkably, the selectivity towards $C_2H_4$ is incredibly high (~99%) among $C_{2+}$ products. There is a small amount of ethanol and negligible acetate in the liquid products and no ethane in the gas products (Supplementary Figs. 37 and 38). The maximum $FE_{C_2H_4}$ and cathodic $EE_{C_2H_4}$ are $84.5 \pm 1.7\%$ and $47.6 \pm 1.0\%$, respectively (at −0.81 V), some of the highest $FE_{C_2H_4}$ and cathodic $EE_{C_2H_4}$ values ever reported in the literature (Supplementary Table 2). The active $C_2H_4$ production is attributed to the stable nanostructure and the Cu/Cu⁺ interfaces (Supplementary Figs. 40 and 41). Moreover, the excellent $C_2H_4$ selectivity does not compromise the current densities. The $C_2H_4$ partial current densities are $92.5$ mA cm⁻² at −0.81 V and $175.2$ mA cm⁻² at −1.01 V, respectively, indicating the high intrinsic CO₂RR activity of the DVL-Cu@GDL catalyst (Supplementary Fig. 42). By comparison, $H_2$ production, the dominant by-product, is severely suppressed in the flow cell, with a FE of $12.6 \pm 1.3\%$ ($13.8$ mA cm⁻²) at −0.81 V, probably due to the high local pH generated by fast proton consumption and the low buffering capability of the KCl electrolyte.

The CO₂RR performance of the two-electrode flow cell configuration was also tested (Fig. 5c and Supplementary Fig. 31). With increasing current density, the voltage increases linearly between −2 and −5 V. At the optimal current density of 75 mA cm⁻², the 77.3% $FE_{C_2H_4}$ coupled with a full-cell voltage of −3.1 V realized an $EE_{C_2H_4}$ of $28.9 \pm 1.3\%$, the highest value achieved in the neutral catholyte in the literature. Surprisingly, the DVL-Cu@GDL catalyst presented an average $FE_{C_2H_4}$ of 74.0% at a constant current density of 150 mA cm⁻² for ~55 h of stable electrolysis without any surface hydrophobic treatment (Fig. 5e and Supplementary Fig. 39). The compact evaporated Cu film and stable nanostructure might have the ability to prevent the GDL from flooding during electrolysis, which accounts for the excellent long-term stability in the flow cell. Furthermore, to improve the $EE_{C_2H_4}$ at a high current density >200 mA cm⁻², we tested our catalyst in a membrane electrode assembly (MEA) electrolyzer (Fig. 5d and f, Supplementary Fig. 43). The full-cell $EE_{C_2H_4}$ increased markedly to $27.6 \pm 0.8\%$ at 200 mA cm⁻² and $23.7 \pm 1.1\%$ at 250 mA cm⁻² using 0.5 M KHCO₃ as the anolyte, with competitive $FE_{C_2H_4}$ values of $80.0 \pm 2.2\%$ and $71.2 \pm 3.3\%$, respectively. In conclusion, the DVL-Cu catalyst realized cost-effective and stable ethylene conversion at a competitive current density.

**Mechanism analysis.** To unravel the intrinsic origin of the high structural stability of DVL-Cu, the electrode current density and electrolyte current density of DVL-Cu and Cu₂O nanocubes (12 nm in length) during the CO₂RR were acquired by COMSOL multiphysics simulations (Fig. 6). Cu₂O nanocubes were chosen as a comparison to show the agglomeration effect, and DVL-Cu carefully preserved its nanostructures after the CO₂RR process (Supplementary Fig. 22). Under an identical total circuit current (100 mA cm⁻²), Cu₂O nanocubes deliver a nearly 5-fold higher

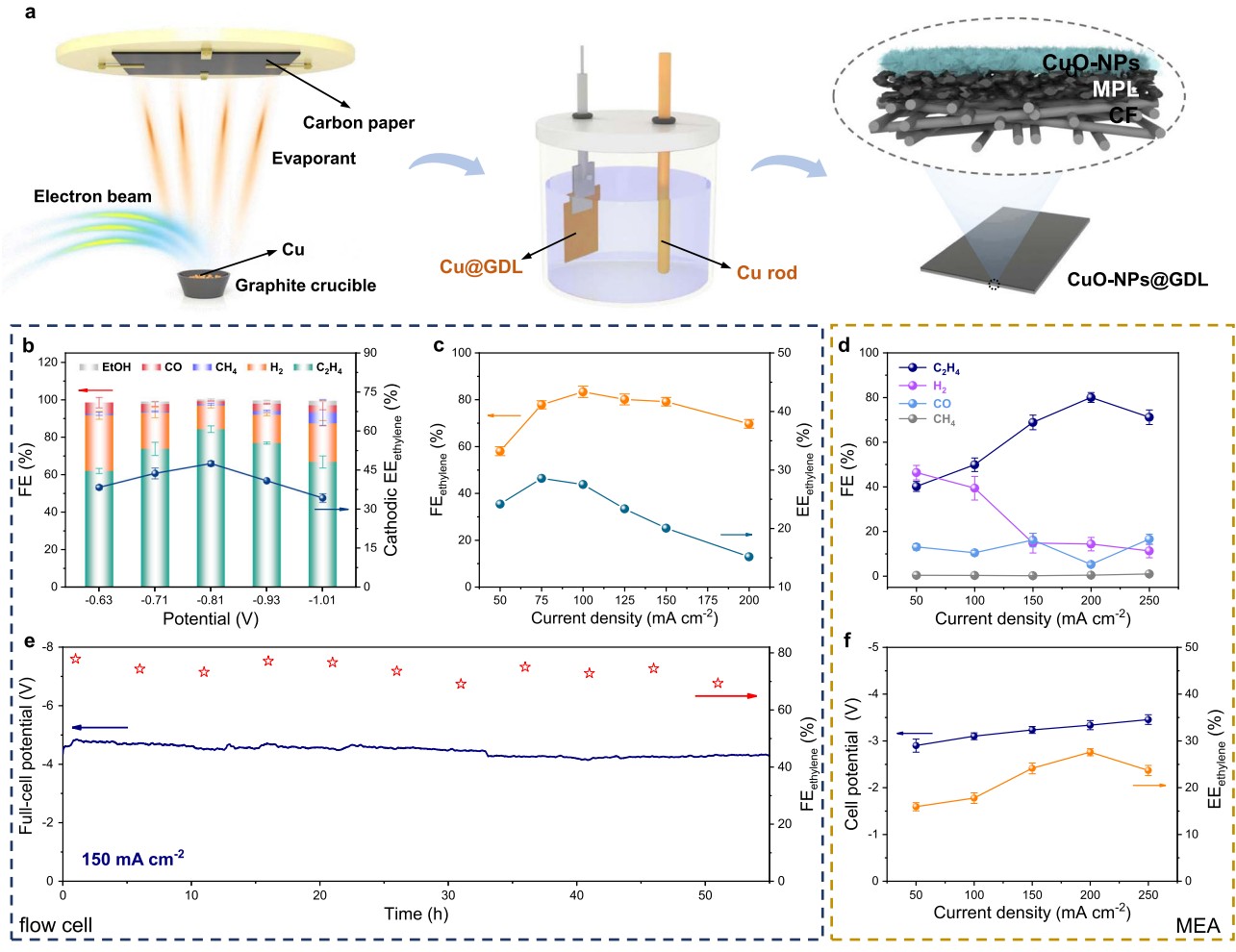

**Fig. 5 Preparation and CO₂RR performance of DVL-Cu@GDL in the flow cell and MEA electrolyzer. a** Schematic illustration of the fabrication of CuO-NPs@GDL. (MPL: microporous layer; CF: carbon fiber) **b** FEs and cathodic $EE_{C_2H_4}$ of the DVL-Cu@GDL catalyst in the flow cell. **c** Full-cell potential and cathodic $EE_{C_2H_4}$ of the DVL-Cu@GDL catalyst in the flow cell. **d** Gas product FEs of the DVL-Cu@GDL catalyst in an MEA electrolyzer. **e** Stability test of the DVL-Cu catalyst at a constant current density of 150 mA cm$^{-2}$ in the flow cell. **f** Cell potential and $EE_{C_2H_4}$ of the DVL-Cu@GDL catalyst in an MEA electrolyzer. Error bars represent the standard deviation of three independent measurements.

electrode current density than DVL-Cu (Fig. 6c, d). Uniformly distributed nanoplates with a large surface area of DVL-Cu effectively disperse the current density, which guarantees structural stability during the CO₂RR. Moreover, the electrolyte current density (correlated to local electrostatic intensity) in the Cu₂O nanocube system is more nonuniform than that in the DVL-Cu system and is especially higher at the corners of the nanocubes (Fig. 6a, b). A faster dissolution/redeposition process occurs in those regions with a higher electrolyte current density, and increased local electrostatic intensity leads to easier electromigration of nanostructures, which results in eventual agglomeration of Cu₂O nanocubes. Simultaneously, the higher electrode current density of Cu₂O nanocubes accelerates this agglomeration process. These simulation results indicate that a moderate electrode current density coupled with the uniformly distributed electrolyte current density of DVL-Cu guarantees its prolonged structural stability during the CO₂RR.

The role of local pH and Cl⁻ in the catalytic performance was investigated by electrolyte analysis. KCl (non-buffering electrolyte), KHCO₃ (buffering electrolyte) and K₂SO₄ (non-buffering electrolyte without Cl⁻) were chosen for comparison. A higher local pH was generated when non-buffering electrolytes were used as catholytes due to their poor pH stabilization capability. By observing the overall product distribution, it is found that the

ethylene production in the presence of KHCO₃ is not satisfactory, with significantly higher methane production (Supplementary Fig. 23, detailed analyses are shown in Note 1). It is suggested that the lower local pH in KHCO₃ electrolyte than that in KCl electrolyte results in inferior ethylene production because lower pH regions favor hydrogen evolution and methane production, both needing H⁺[50]. After comparing the catalytic performance in KCl and K₂SO₄, we conclude that the existence of Cl⁻ suppresses hydrogen evolution, especially at high overpotentials, which is considered the result of the Cl⁻-specific adsorption. The strongly adsorbed Cl⁻ facilitates electron transfer from the electrode to CO₂ and suppresses the adsorption of protons, leading to a higher hydrogen evolution overpotential[51].

The long-term stability test results in different electrolytes are also quite informative. Interestingly, DVL-Cu delivered an ~50 h stable current density and ethylene FE in K₂SO₄ under −0.8 V. In comparison, it only preserved stable performance for less than 10 h in KHCO₃ under the same conditions. SEM and TEM images of the post-CO₂RR sample in K₂SO₄ (Supplementary Fig. 24) display morphology and Cu(0)/Cu(I) interfaces equivalent to those obtained from the KCl sample, indicating that a high local pH plays a critical role in the prolonged stability, rather than morphology. Detailed post-electrolysis characterizations were performed to determine the stability origin for the samples obtained

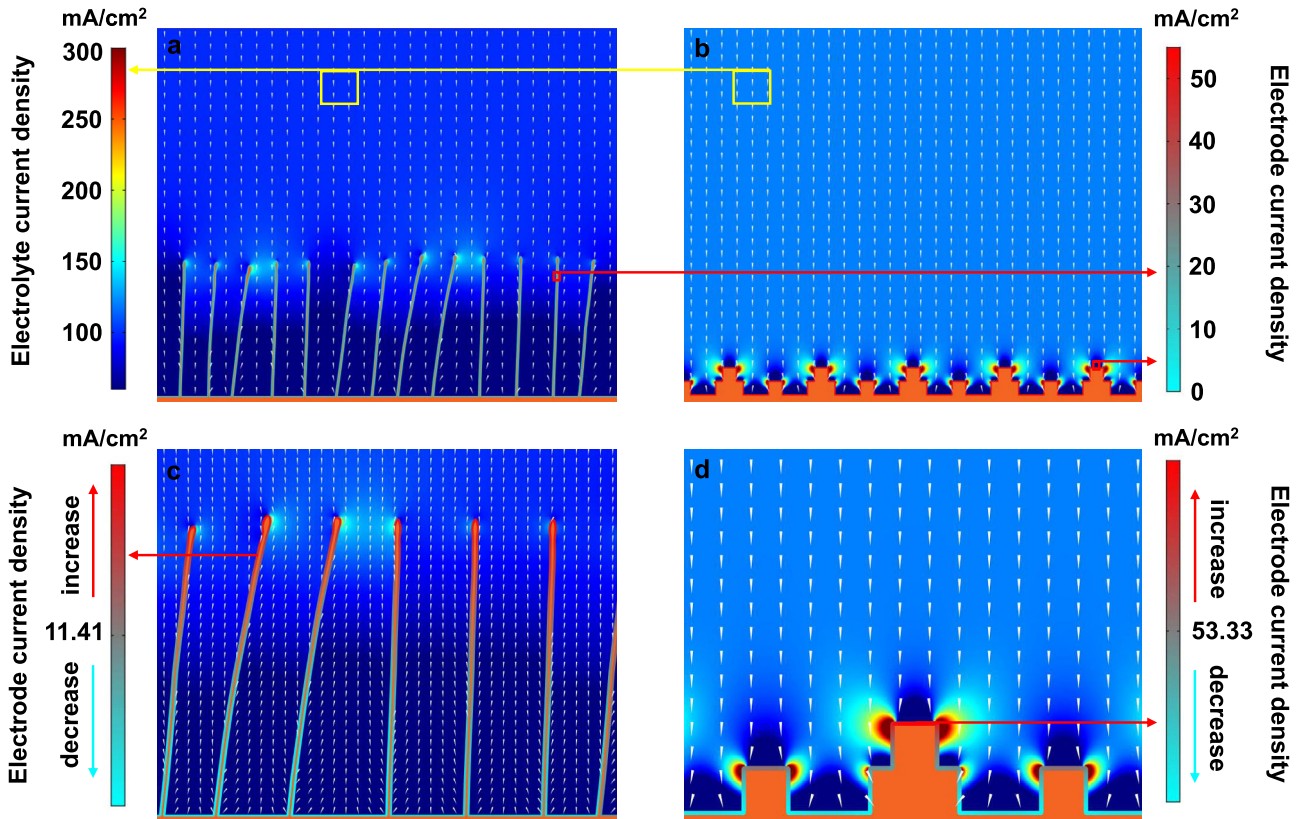

**Fig. 6 COMSOL multiphysics simulations. a, c** Electrode current density and electrolyte current density distribution of DVL-Cu at 100 mA cm$^{-2}$. **b, d** Electrode current density and electrolyte current density distribution of Cu$_2$O nanocubes at 100 mA cm$^{-2}$. The electrode current density and electrolyte current density (in **a** and **b**) correspond to the left and right legends, respectively. The electrode current densities in (**c**) and (**d**) correspond to the left and right legends, respectively. The conical arrows represent the electrolyte current density vector.

from KHCO$_3$ and KCl. EELS mapping of the KHCO$_3$ sample (Supplementary Fig. 26) reveals that Cu(I) species agglomerate on the top of nanoplates, while these species distribute uniformly in the KCl sample (Supplementary Figs. 27 and 28). AES depth profile analyses (Supplementary Fig. 29) further verify the Cu(I) species distribution differences, where the Cu(I) content in the KCl sample is higher with depth than that in the KHCO$_3$ sample. Apparently, non-buffering electrolytes stabilize the DVL-Cu catalyst by protecting its Cu(0)/Cu(I) interfaces during electrolysis. Since the K$_{sp}$ of CuOH is relatively low (1.0*10$^{-14}$), a high local pH significantly slows the dissolution of Cu(I) species. Hence, a high local pH suppresses the dissolution/redeposition of Cu(I) species in non-buffering electrolytes, which preserves the Cu(0)/Cu(I) interfaces during the CO$_2$RR.

To gain mechanistic insights into the CO$_2$RR catalytically active sites and reaction pathway of DVL-Cu, we further performed DFT calculations to investigate the energy barrier for the production of C$_2$H$_4$ and other products at different Cu-containing sites. Here, Cu(110) and Cu$_2$O(110) slabs were first constructed as the model of Cu$^0$ and Cu$^+$ catalytic sites based on the STEM and GI-XRD results, and Cu(110) was considered the corresponding active plane to produce C$_{2+}$ products in reference[25]. The Cu(110)/Cu$_2$O(110) interface was modeled by distributing the Cu$_2$O(110) slab on the surface of the Cu(110) slab (Supplementary Figs. 44–46, the size effect of Cu$_2$O slab is discussed in Supplementary Fig. 47, Supplementary Table 1 and Supplementary Note 3). Then, the energy barrier of each step in the reaction pathway on the Cu(110) slab, Cu$_2$O(110) slab and Cu(110)/Cu$_2$O(110) interface was calculated to evaluate the catalytic performance of different catalytic sites in DVL-Cu.

First, the C$_2$H$_4$ formation pathway through the *CO–*COH dimerization pathway was investigated to reveal catalytic sites on DVL-Cu (Supplementary Tables 4 and 5). The *CO–*COH dimerization pathway was confirmed to be most favorable over other pathways considering the reaction energy and kinetic adsorption structure (Fig. 7a and Supplementary Figs. 48–50). The reduction of *CO$_2$ to *COOH occurs with $\Delta G = 0.50$ eV on Cu(110)/Cu$_2$O(110), which is lower than that on Cu(110) ($\Delta G = 0.76$ eV) and Cu$_2$O ($\Delta G = 1.05$ eV) since the interfaces promote the stabilization of the *COOH intermediate. Therefore, Cu(110)/Cu$_2$O(110) secures a higher surface *CO coverage, which is consistent with the higher FE$_{CO}$ of DVL-Cu at a low overpotential. More importantly, the energy barriers for the rate-determining step (RDS) are 0.60 eV (*CO + *CO → *CO + *COH), 1.19 eV (*CO + *CO → *CO + *COH) and 1.05 eV (*OH$_2$CCH → *O + C$_2$H$_4$) on Cu(110)/Cu$_2$O(110), Cu(110) and Cu$_2$O(110), respectively (Fig. 7b, c). Hence, C$_2$H$_4$ production happens more easily on the Cu(110)/Cu$_2$O(110) interface, indicating that these interfaces are the catalytically active sites in DVL-Cu. Furthermore, DFT calculations reveal that the adsorption energy of the *OCCOH intermediate is −0.77 eV on Cu(110)/Cu$_2$O(110), which is more negative than that on Cu(110) (−0.18 eV) and Cu$_2$O(110) (−0.28 eV) (Fig. 7c). These results verify that the presence of heterointerfaces are beneficial to the adsorption of the post-dimerization intermediate (*OCCOH), thus reducing the energy barrier of C–C dimerization. The carbonyl C binding to the interface while the hydroxyl C binding to Cu atoms is the optimized adsorbed structure of the *OCCOH intermediate on the Cu(110)/Cu$_2$O(110) interface (Fig. 7d). Moreover, C$_2$H$_4$ desorption is important for the regeneration of

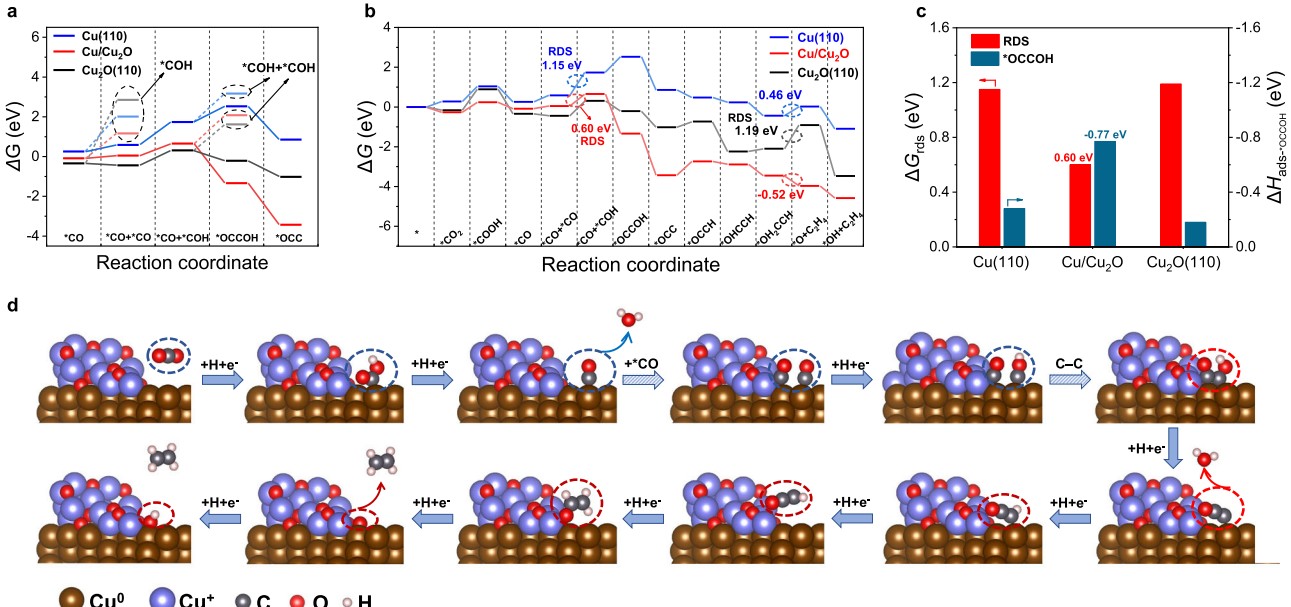

**Fig. 7 DFT calculation results. a** A reaction energy diagram for the $CO_2RR$ to *COH (the intermediate of $CH_4$ formation) and to *OCC through different dimerization pathways on the Cu(110) slab, $Cu/Cu_2O$ interface and $Cu_2O(110)$ slab. **b** A reaction energy diagram for the $CO_2RR$ to $C_2H_4$ on the Cu(110) slab, $Cu/Cu_2O$ interface and $Cu_2O(110)$ slab. **c** The free energy of the rate-determined step for the $CO_2RR$ to $C_2H_4$ pathway and the adsorption energy of the *OCCOH intermediate on the Cu(110) slab, $Cu/Cu_2O$ interface and $Cu_2O(110)$ slab. **d** Optimized structures for the reaction intermediates of the $C_2H_4$ formation pathway on the $Cu/Cu_2O$ interface.

active sites, which is crucial for the production rate of $C_2H_4$. Interestingly, this step is exergonic only on Cu(110)/$Cu_2O(110)$ (−0.52 eV), indicating that DVL-Cu possesses a fast $C_2H_4$ production capacity.

The high selectivity towards $C_2H_4$ was then studied by comparing the free energy barriers of different catalytic products. The formation of $CH_4$ via different pathways was primarily considered. As shown in Supplementary Fig. 51, $CH_4$ production related to the *HCOO intermediate is energetically less favorable than other pathways (leading to $CH_4$ or other products) related to the COOH* intermediate in the first hydrogenation step on Cu(110)/$Cu_2O(110)$. Based on the *COOH intermediate, the free energy barriers of the second hydrogenation step are 1.57 eV (*HCOOH) and −0.32 eV (*CO) on the $Cu/Cu_2O$ interface, respectively, which indicates that *CO tends to form more than the *HCOOH intermediate. Afterwards, insurmountable energy barriers are required for the hydrogenation of *CO to form *COH on three catalytic sites, while the free energy required for another *CO adsorption step is lower than that for *CO hydrogenation on all three sites, making the reaction pathway to $C_{2+}$ products possible. These factors account for the high $C_2H_4/CH_4$ product ratios of the DVL-Cu catalyst. Otherwise, the energy barriers required for $H_2$, formate and $C_2H_5OH$ are dramatically larger than that for $C_2H_4$ (Supplementary Figs. 51–53). Hence, the formation of $C_2H_4$ is energetically more favorable than other products, as demonstrated by DFT calculations, which is identical to the $CO_2RR$ product analysis results.

In brief, the adsorption of two *CO species, followed by hydrogenation of one of the *CO species and the consecutive dimerization of *CO and *COH to form *OCCOH, is considered the most favorable pathway for $C_2H_4$ production on the Cu(110)/$Cu_2O(110)$ interface in DVL-Cu. The insurmountable energy barrier for the hydrogenation of a single adsorbed *CO and the facile $C_2H_4$ formation pathway on the Cu(110)/$Cu_2O(110)$ interface result in the high $C_2H_4/CH_4$ ratio in $CO_2RR$ products. These DFT results suggest that the existence of $Cu/Cu_2O$

interfaces reduces the energy barrier of C–C dimerization and accelerates the desorption of $C_2H_4$, leading to highly active and selective $C_2H_4$ production on DVL-Cu.

## Discussion
In summary, we have proposed an anodic oxidation method for the large-scale preparation of oxide-derived Cu catalysts with stable $Cu/Cu_2O$ interfaces for highly active $CO_2RR$ to $C_2H_4$ with high FE and prolonged stability. The high oxidation degree of Cu foil with vertically arranged Cu nanoplates prevents the agglomeration of nanostructures and preserves stable $Cu/Cu_2O$ interfaces during the $CO_2RR$. Utilizing these advantages, the DVL-Cu catalyst achieves a high $FE_{C_2H_4}$ of 84.5 ± 1.7% and $EE_{C_2H_4}$ of 28.9 ± 1.3% in the flow cell and 27.6 ± 0.8% $EE_{C_2H_4}$ at 200 mA cm$^{-2}$ in the MEA electrolyzer. Moreover, the DVL-Cu catalyst maintains consistent electrolysis performance for ~55 h in the flow-cell. Mechanism analysis indicates that a moderate electrode current density and uniform electrolyte current density coupled with high local pH guarantee structural and interfacial stability, while Cl$^−$-specific adsorption suppresses hydrogen evolution at higher overpotentials. DFT calculations reveal that the energy barrier for C–C coupling is significantly reduced because Cu$^+$ species enhance the adsorption capacity of the *OCCOH intermediate. The good selectivity, prolonged stability and facile production of the DVL-Cu catalyst highlight its application potential in realizing the industrial conversion of $CO_2$ to $C_2H_4$.

## Methods
**DFT calculations**. All DFT calculations were performed using Vienna ab initio simulation package (VASP)[52], within the projector-augmented wave (PAW) potentials[53] together with the generalized gradient approximation (GGA) exchange-correlation[54] proposed by Perdew–Burke–Ernzerhof (PBE)[55] to calculate the correlation energies. The bulk-unit cells for pure Cu and $Cu_2O$ were constructed and the k-mesh were 19 × 19 × 19 and 7 × 7 × 10, respectively. The Cu(110) surface, a main exposed facet in the experimental result, was composed of four layers with 3 × 3 supercells, and the $Cu_2O(110)$ supported on Cu(110) surface was modeled by adding $Cu_2O(110)$ clusters of 8 $Cu_2O$ to the Cu(110) surface. The

bottom two layers of Cu(110) were fixed and the vacuum space was set as 20 Å to avoid interactions with their periodic images. The $3 \times 5 \times 1$ Monkhorst–Pack k-point meshes and plane-wave cutoff energy of 520 eV were used in all calculations. The convergence tolerances for residual force and energy were set to 0.01 eV Å$^{-1}$ and $10^{-5}$ eV, respectively.

**Preparation of the catalysts.** For the DVL-Cu catalyst used in the H-cell, Cu foils (Alfa Aesar, 0.025 mm, 99.8%) were cut into $1.5 \times 3$ cm$^2$ and annealed at 1050 °C for 3 h under Ar/H$_2$ atmosphere to remove the copper oxide layer on it. Then the Cu foil was fixed onto a platinum electrode clip to form a working electrode. The counter electrode was a Cu rod made from curled Cu foil. The electrolyte used for anodic oxidation was 1.0 M NaOH (Alfa Aesar, 98%). Galvanostatic oxidation on the Cu foil at a constant current density of 0.26 mA cm$^{-2}$ was performed by a CHI 760E potentiostat until the surface of Cu foil entirely turned into black. The CuO-NPs were washed five times by deionized water before being used as the CO$_2$RR catalyst. For the R-CuO-NPs catalyst, the CuO-NPs were annealed at 450 °C for 3 h under Ar/H$_2$ atmosphere.

For the DVL-Cu@GDL catalyst used in the flow cell, a 500 nm thick Cu layer was primarily deposited on the carbon paper (Sigracet 28 BC) by an electron beam evaporation system (DZS500, SKY). During the evaporation process, 100 g Cu particles were placed in a graphite crucible inside the evaporation chamber. The chamber pressure was vacuumized to $10^{-6}$ Torr by the molecular pump. A thin Cu layer was deposited on carbon paper at an evaporation rate of 1 Å s$^{-1}$ controlled by the film thickness measurement system. GDL was kept rotating at a slow speed of 50 rpm during evaporation. Evaporated carbon paper was fixed onto a platinum electrode clip to form a working electrode. The electrochemical oxidation process was the same as the Cu foil except that the current density was 0.13 mA cm$^{-2}$ for the carbon paper.

**Materials characterization.** SEM images were taken by a ZEISS SUPRA55 microscope. A JEOL F200 microscope was used to take the TEM images. AFM images were obtained by Bruker Dimension FastScan microscope. Aberration corrected STEM imaging and EELS mapping were acquired from a Nion HERMES-100 under 100 kV with a 30 mrad convergence angle. The enlarged STEM-BF image is denoised by low-psss filtering. Cu valence state analysis was performed by multiple linear least squares (MLLS) fitting in the 920–960 eV energy-loss range. The processed EELS data has been calibrated along the energy-loss axis to much the standard data[56], as the as-acquired spectra deviate slightly due to the small non-linearity of the energy dispersion at the two ends of the spectrometer prism. XPS spectra (ESCALAB 250Xi, Thermo Fisher Scientific Inc., USA) was used to investigate chemical compositions and elemental oxidation states of the catalysts. Raman spectra were obtained from the Raman spectrometer (Horiba, Olympus microscope) with a 532 nm laser. GI-XRD patterns were obtained by a Panalytical Empyrean X-ray diffractometer. Gas products were analyzed by a Shimadzu GC 2030 gas chromatograph. Liquid products were analyzed by a NMR spectroscopy (AVANCE III 600 M, Bruker).

**Electrochemical CO$_2$ reduction measurements.** A gas-tight electrolysis H-cell (Gaoss Union, 50 mL) separated by a Nafion 117 membrane (Sigma Aldrich) was used to measure the CO$_2$RR performance of the catalysts. 30 mL 0.5 M KCl (Sigma Aldrich) was employed as catholyte and anolyte. Before CO$_2$RR test, 500 standard cubic centimeters per minute (sccm) CO$_2$ (99.999%, Praxair) was constantly bubbled into the electrolytes for 30 min to saturated it with CO$_2$. A Pt foil (Pine Instruments, $1 \times 1$ cm$^2$) and a Ag/AgCl electrode (Gaoss Union) filled with saturated KCl solution were used as counter electrode and reference electrode, respectively. We kept the CO$_2$ flow rate at 20 sccm and 1000 rpm stirring of the catholyte during the CO$_2$RR. For the long-term CO$_2$RR stability test, the CO$_2$-saturated KCl electrolytes were replaced every 12 h. The electrolysis was performed by a CHI 760E potentiostat using chronoamperometry method at each applied potential for 1 h to measure the FE of each product. All applied potentials were converted to the RHE by the equation: $E$ (vs. RHE) = $E$ (vs. Ag/AgCl) + 0.204 V + 0.0591 V × pH − $iR$, with $iR$ compensation. The gas products combined with CO$_2$ gas were injected into a six-way valve, which is linked with an online GC-BID. The gas chromatography was calibrated by five standard gases (H$_2$, CO, CH$_4$, C$_2$H$_2$, C$_2$H$_4$ and C$_2$H$_6$ in CO$_2$) with gradient concentrations at 20 sccm flow rate before using. The gas samples were analyzed at least after 30 min electrolysis to insure the CO$_2$RR reaching a stable state. Liquid samples were collected after 1 h electrolysis and measured by $^1$H NMR with dimethyl sulfoxide (DMSO) as an internal standard. The Faradaic efficiencies (FEs) were calculated on the basic of the following equation:

$$\text{Faradaic effciencies} = \frac{Q_x}{Q_{total}} = \frac{n_x N_x F}{Q_{total}} \quad (1)$$

where $Q_x$ and $Q_{total}$ was the charge passed into product $x$ and totally passed charge (C) during CO$_2$RR, $n_x$ represents the electron transfer number of product $x$, $N_x$ was the product amount (mol) of $x$ measured by GC or NMR and $F$ was the Faraday constant (96485 C mol$^{-1}$).

The (cathodic) energy efficiencies were calculated on the basic of the following equation:

$$\text{energy effciencies} = \frac{E^{\ominus}}{E_{applied}} \times \text{FE}_{C_2H_4} \quad (2)$$

$$\text{cathodic energy effciencies} = \frac{E^{\ominus}}{1.23\,\text{V} - E_{applied}} \times \text{FE}_{C_2H_4} \quad (3)$$

where $E^{\ominus}$ is the thermodynamic potential for the ethylene formation (1.15 V), $E_{applied}$ represents the potential applied during the CO$_2$RR.

For the electrochemical CO$_2$RR test in a flow cell, a commercial flow cell electrolyzer (Gaoss Union, 1 cm$^2$ active area) was used. The CuO-NPs@GDL was placed between the gas chamber and catholyte chamber, and the catholyte and anolyte chambers were separated by an anion-exchange membrane (FAA-3-PK-130, Fumapem). A Ag/AgCl (filled with saturated KCl solution) electrode and a Ni foam with NiFe hydroxides deposited on it using a electrodeposition method reported previously[57] were employed as reference electrode and counter electrode, respectively. KCl and KOH solution were served as the catholyte and anolyte, respectively. The gas flow rate was 50 sccm during CO$_2$RR and the gas products were injected into a six-way valve. The catholyte and anolyte were circulated by a peristaltic pump at 10 sccm and 200 sccm, respectively. The electrolysis of the flow cell was performed on a potentiostat (CS-150CN, CorrTest) equipped with a 2 A current booster.

For the electrochemical CO$_2$RR test in an MEA electrolyzer, a commercial MEA electrolyzer (Shanghaikeqi, 5 cm$^2$ active area) was used. The CuO-NPs@GDL, an anion-exchange membrane (Sustainion® X37-50) and Ti-IrO$_2$ mesh were compressed to form MEA. The MEA was placed between the anode chamber and cathode chamber, then assembled together using associated bolts. 0.5 M KHCO$_3$ solution was served as the catholyte. The gas flow rate was 50 sccm during CO$_2$RR and the gas products were injected into a six-way valve. The catholyt was circulated by a peristaltic pump at 200 sccm. The electrolysis of the MEA electrolyzer was performed on a potentiostat (HSP-3010, Henghui).

**ECSA measurements.** The electrochemical double-layer capacitance method was used for ECSA measurements. In a typical procedure, the catalysts were reduced at −0.6 V vs. RHE for 2 min and cyclic voltammograms at different scan rates (20, 40, 60, 80 and 100 mV s$^{-1}$) have been obtained in Ar-saturated 1 M KOH solution in the non-Faradaic region (−0.07 to 0.13 V vs. RHE) when the curves of different cycles overlapping. The difference between anodic current and cathodic current at 0.03 V of different scan rates was recorded and plot against the scan rates. The 1/2 slope values of these curves were calculated as the double-layer capacitance for corresponding catalysts. The ratios of the double-layer capacitance of the catalysts versus the electropolished copper foil were calculated as the RF[58].

**COMSOL multiphysics simulations.** A two-dimensional finite element model was developed to describe the difference of current density distribution between plate electrode and square electrode. A two-dimensional cross-section of $1000 \times 800$ nm near the electrode was taken for the computational domain. The slice electrodes are represented by thin irregular columns, while the cube electrodes are assumed to be stacked squares. The average height of the sheet electrode is about 300 nm, and the square electrode is about 50 nm.

The tertiary current distribution module from the COMSOL multiphysics software was employed for the finite element simulation. The transport of the K$^+$ and the anion was solved by the Nernst-Plank equation:

$$\nabla \cdot \left( D \nabla c_i + \frac{D z_i e}{k_B T} c_i \nabla V \right) = 0 \quad (4)$$

Where $c_i$ is the concentrations of the potassium or anion ion, $z_i$ are the valences of ions, $e$ is the elementary charge, $k_B$ is Boltzmann constant, the absolute temperature $T$ is set as 297.3 K. The reaction current density was obtained by the Butler–Volmer equation:

$$i = i_0 \left[ \exp\left( \frac{\alpha_a n F \eta}{RT} \right) - \exp\left( -\frac{\alpha_c n F \eta}{RT} \right) \right] \quad (5)$$

Where the $\alpha_a$ and $\alpha_c$ are the dimensionless anodic and cathodic charge transfer coefficients, respectively. $n$ is the number of electrons involved in the electrode reaction, $R$ is the universal gas constant, $F$ is the Faraday constant. The exchange current density $i_0$ was obtained by the Arrhenius law:

$$i_0 \propto \exp\left( -\frac{E_a}{k_B T} \right) \quad (6)$$

Where the $E_a$ is the activation energy of the reaction, which was experimentally obtained to be 0.21 eV with K$^+$ involved.

Boundary conditions: The upper boundary was set as the electrolyte boundary with a current density at 100 mA/cm$^2$, and an ion and anion concentration with 0.5 mol/L. The electrolyte conductivity was assumed to be 10 S/m. The diffusion coefficient of Li$^+$ and the anion were set as $5.273 \times 10^{-9}$ m$^2$/s and $2.032 \times 10^{-9}$ m$^2$/s, respectively.

## Data availability
The data supporting this study are available within the paper and the Supplementary Information. All other relevant source data are available from the corresponding author upon reasonable request.

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

## Acknowledgements

This work was supported by the National Key R&D Program of China (Grant No. 2018YFA0306900), the National Natural Science Foundation of China (No. 51872012), the Fundamental Research Funds for the Central Universities and the 111 Project (B17002).

## Author contributions

These authors contributed equally: W.L., P.Z., A.L., and B.W. Y.G. and W.Z. conceived the project and supervised the research work. W.L. and P.Z. designed and conducted most of the experiments of this project. A.L. and W.Z. performed the STEM measurements and analyzed the results. B.W. and R.Z. designed the computational studies and analyzed the computational data. G.Z. conducted the GI-XRD measurements. K.S., X.G., and Q.C. performed the electronic beam evaporation. W.L., Y.W. and X.W. co-wrote the manuscript. Y.G., W.Z., and R.Z. discussed the results and reviewed the manuscript.

## Competing interests

The authors declare no competing interests.
