## [Peer review file · Nature Communications]

REVIEWER COMMENTS

Reviewer #1 (Remarks to the Author):

This manuscript employed an oxygen-rich ultrathin CuO nanoplates arrays, in which Cu/Cu₂O heterogeneous interfaces were formed, for electrochemical CO₂ reduction reaction (CO₂RR) process. As presented in the manuscript, the performance was good (i.e., 84.5% selectivity toward C₂H₄ at -0.8 V with a current density of ~ 100 mA cm⁻²), and the Cu(0)/Cu(I) interfaces related structure-activity relationship was proposed. It was suggested by authors that the insurmountable energy barrier for the hydrogenation of single adsorbed *CO and the facile C₂H₄ formation pathway on the Cu(110)/Cu₂O(110) was the possible reason accounting for superior C₂H₄ selectivity. However, from the perspective of internal mechanism, research novelty, and methodology, this work still does not meet the similar level with recent alike publications in Nature Communications. Detailed comments are given as following:

1. The authors reported that oxygen-rich ultrathin CuO nanoplates arrays delivered a high selectivity toward C₂H₄ in CO₂RR, and the authors concluded the mechanism to be a Cu(0)/Cu(I) interfaces effect. However, similar researches on Cu/Cu₂O or Cu(0)/Cu(I) have been extensively reported [e.g., Altaf N et al., Journal of Energy Chemistry, 2020, 48: 169-180; Yan Z et al. Catalysis Science & Technology, 2021]. Therefore, it is not enough to supply a simple DFT calculation, as well as a series of experiments for confirming the formation of copper/cuprous oxide interfaces, for supporting the authors' viewpoint. The authors should clearly describe the differences between this work and previous studies, instead of simply listing the comparison of the CO₂RR performances.
2. Cu-based catalysts have been widely employed in CO₂-to-C₂H₄ conversion so far, so I would be curious what are the intrinsic differences between this CuO system and previous reports [e.g., Dinh C T et al., Science, 2018, 360(6390): 783-787.] Indeed, the manuscript at present status cannot supply a solid explanation from the microscopic mechanism. In-depth research in this field should be necessary for publishing this work in Nature Communications.
3. According to the Abstract in Line 38 and Line 87, the best performance was achieved in neutral KCl electrolyte, which may be more interesting than the performance data. As most of the reported highest performances were obtained in alkaline systems, which typically suffer from a severe carbonate side reaction. A well-established system in neutral electrolyte may be an important progress toward practical application of CO₂RR. But the authors only explained why KCl was chosen at present. The authors should provide a more detailed reason, as well as demonstrate a convincing mechanism on this phenomenon.

4. As indicated in the article's title, the long-term stability is important for this work's novelty. However, the authors didn't give a critical analysis from the performance. For instance, why Cu(0)/Cu(I) interfaces, a thermodynamic factor according to authors' DFT calculations, could result in such a stable activity (i.e., dynamic phenomenon), which has not been reported in previous Cu/Cu₂O studies?

5. To follow up the previous question, what were the catalyst structures after the 55 hours of flow cell tests? Could these Cu(0)/Cu(I) interfaces be maintained after this long-term test at 150 mA cm⁻²? The authors should provide detailed post-electrolysis characterizations and give a clear explanation. Besides, EIS cannot supply much useful information in such a multistep reaction with unknown pathways.

6. In the Supporting information, the authors presented the 3D optical profiler simulated images and calculated results of CuO; and in Line 128, the authors stressed that "The Ra of CuO-NPs is 128 5.96 μm, dramatically larger than that of CuO (3.730 μm)". Later in the text (Line 223), roughness factor (RF) was also calculated by double-layer capacitance method. So, what is the difference between ECSA and roughness factor? And what can these data be useful for the CO₂RR evaluation?

Reviewer #2 (Remarks to the Author):

This paper presents a combined experimental and DFT study of CO₂ electrochemical conversion to C₂ products on CuO/Cu catalysts. The authors need to address the following points before it is considered as a potential Nat. Commun publication:

(i) The authors claimed that C₂H₄ is the main product. However, Figure 4e clearly indicates that ethanol is the main product. The same figure also shows that formate is one of the main products. There is no C₂H₄ in Figure 4c.

(ii) As the authors mentioned HER is always a competing reactions. The HER free energy diagram is not included and discussed in the manuscript. The authors should perform DFT to get HER energetics and should be compared with the CO₂RR free energy diagrams to show that HER is less favorable and the catalysts selectively promote the CO₂RR.

(iii) As shown in Figure 4c, the authors should also include and discuss the formate pathway and compared with the HOCO pathway which is claimed to produce C₂ products.

(iv) Since ethanol is one of the main C₂ products, it would be helpful to have free energy diagram that leads to the formation of C₂H₅OH and be compared with the C₂H₄ pathway.

(v) The authors assumed that CH₄ forms from COH intermediate. However there are several other channel that lead to the formation of CH₄. For example: CO₂==>HCOO==>HCOOH==> H₂COOH==<

$\text{H}_2\text{CO} \Rightarrow \text{H}_3\text{CO} > \text{CH}_3 \Rightarrow \text{CH}_4$. There is no rationale presented why other channels are not included in the discussion.

(vi) The selection of DFT model for the interface needs to be justified. Did authors test other models (smaller/larger nanoparticle)? What is the size of Cu₂O nanoparticle in experiment? Does the DFT model correspond to experiments?

Reviewer #3 (Remarks to the Author):

Wei Liu et al., presented a facile synthesis method to prepare Cu-based electrocatalyst for the CO₂RR. They have shown that their catalyst, called dense vertical lamellate Cu nanoparticles (DVL-Cu), under the reduction applied potential maintains the Cu₂O/Cu interface to keep ethylene Faradaic efficiency (FE) at high value for a long time, while with a considerably improved total energy efficiency (EE). The attained FE for ethylene is 84.5%, and stable for 55 hours, which is a significant improvement compared to the previous reports so far. A thorough set of characterizations is performed to shed light on the relation between catalyst composition/structure and its superior activity. Overall, this reviewer is confident to recommend the manuscript to be published in the journal Nature Communications, after considering the following suggestions.

In page 12, line 252, it is recommended to add a SEM image of DVL-Cu after 2h CO₂RR (and through comparison) showing that the agglomeration of Cu nanoparticles is the main reason for the instability.

The author emphasized the capability of this method for industrialization, yet using KCl as the electrolyte is quite hazardous, and it might be a big challenge for industrialization.

In Fig. 4e, it is recommended to plot FE rather than ratios, albeit the FEs are quite small. Otherwise, this reviewer does not find the figure informative. Also, please use the same potential range for Fig. 4 A and B.

In line 230 and 231, the author mentioned the generation of H₂ is severely suppressed. 21% FE is not considered a "severely" suppression. Also, total FEs are not reported. From the current results it seems the total FE goes above 100%. In addition, the rationale provided in line 232 for the H₂ suppression is not satisfying. This reviewer suggests testing both catalysts for the HER and compare the results. It is possible that the DVL-Cu intrinsically is less active for the HER.

In lines 248-252: why two catalysts are tested at two different potentials for the stability test? They must be compared under similar conditions, otherwise, drawing a conclusion would not be possible. Either test either DVL-Cu at -1.0V or R-CuO-NPs at -0.8V.

Lines 253-261: although Cu₂O was shown in the HRTEM for DVL-Cu, it is recommended adding GI-XRD profiles after 1h CO₂RR, like what has been done for the R-CuO-NPs.

In the table 1, the partial current density of C₂H₄ was mentioned 174.4 mA/cm² which does not seem to be super accurate. At -0.81V the partial current density was mentioned 92.5 mA/cm² (line 289).

Line 299, the author claimed this is the highest value achieved in the “neutral” catholyte. This requires further evidence, e.g., measuring pH, during and (especially after the long stability experiment). A plot of pH vs. time will be very informative.

The manuscript needs another round of proofreading as there are several typos and sentences with grammatical errors which make it difficult to understand. Few examples are mentioned below:

- Line 32: “an facile”
- Fig. 1c: CuO growth and CuO-nucleation graph caption should be revised.
- Supplementary Fig. 10 caption: R-CuO instead of A-CuO. This is repeated many times.
- Line 93: “catholic” reduction
- Supplementary Fig. 20 caption is incorrect.
- Line 150: the authors discuss the decrease in Cu (110) peak intensity but no peak is assigned to this facet in Fig. 2d.
- Fig. 2F: don't we expect to observe an intensified Cu₂O peak at -1V compared to that at -0.6V after 1800 s?

REVIEWER COMMENTS

Reviewer #1 (Remarks to the Author):

This manuscript employed an oxygen-rich ultrathin CuO nanoplates arrays, in which Cu/Cu₂O heterogeneous interfaces were formed, for electrochemical CO₂ reduction reaction (CO₂RR) process. As presented in the manuscript, the performance was good (i.e., 84.5% selectivity toward C₂H₄ at -0.8 V with a current density of ~ 100 mA cm⁻²), and the Cu(0)/Cu(I) interfaces related structure-activity relationship was proposed. It was suggested by authors that the insurmountable energy barrier for the hydrogenation of single adsorbed *CO and the facile C₂H₄ formation pathway on the Cu(110)/Cu₂O(110) was the possible reason accounting for superior C₂H₄ selectivity. However, from the perspective of internal mechanism, research novelty, and methodology, this work still does not meet the similar level with recent alike publications in Nature Communications. Detailed comments are given as following:

1. The authors reported that oxygen-rich ultrathin CuO nanoplates arrays delivered a high selectivity toward C₂H₄ in CO₂RR, and the authors concluded the mechanism to be a Cu(0)/Cu(I) interfaces effect. However, similar researches on Cu/Cu₂O or Cu(0)/Cu(I) have been extensively reported [e.g., Altaf N et al., Journal of Energy Chemistry, 2020, 48: 169-180; Yan Z et al. Catalysis Science & Technology, 2021]. Therefore, it is not enough to supply a simple DFT calculation, as well as a series of experiments for confirming the formation of copper/cuprous oxide interfaces, for supporting the authors' viewpoint. The authors should clearly describe the differences between this work and previous studies, instead of simply listing the comparison of the CO₂RR performances.

We thank the reviewer for this constructive suggestion. To summarize, the main differences between our work and previous studies are the stability of exquisite nanostructures derived from the ultrathin CuO nanoplates arrays, the stability of numerous delicate Cu/Cu₂O interfaces due to utilization of neutral electrolyte and the assistance of chloride ions. Although much attention has been devoted to the Cu(0)/Cu(I) systems for CO₂RR, the unsatisfactory faradaic efficiency and poor stability still exist in these catalysts.

Firstly, those systems developed in references without exquisite nanostructures usually suffer from severe morphology collapse after being reduced, which diminishes the number of active sites and increases the risk of agglomeration during CO₂RR. Agglomeration of catalysts was familiar in previous research [e.g., Altaf N et al., *Journal of Energy Chemistry*, 2020, 48: 169-180; Yan Z et al. *Catalysis Science & Technology*, 2021] and was a principal cause of poor stability. In the two articles mentioned by the reviewer, there may be problems with vague nanostructures. For example, the flower-shaped Cu₅Al-CO₃ precursor severely agglomerated after being reduced to CO₂RR catalyst in **Journal of Energy Chemistry**, 2020, 48: 169-180. In the revised version, we cited these two references and compared them with our results.

To better answer the question raised by the reviewer, we also compared the morphology evolution of CuO ultrathin nanoplates with other samples. As shown in Figs. R1a-c, CuO nanoplates are only a few atomic layers thick, which have numerous defects and undercoordinated sites. The ultrathin nanoplates can easily self-evolved into well-maintained morphology with numerous active sites exposed (Fig. R1d). The morphology and Cu(I)/Cu(0) distribution were carefully investigated after the long-term test, and details are shown in comment 3 (Figs. R8, 10 and 11). The well-maintained delicate morphology during precursor reduction and CO₂RR process guarantees superior catalytic performance and stability of DVL-Cu. The morphologies of Cu₂O nanocubes and thermal oxidation derived CuO before and after CO₂RR were characterized to display the agglomeration of these unstable nanostructures intuitively. As shown in Figs. R2 and R3, Cu₂O nanocubes severely agglomerate after 1h CO₂RR while the thermal oxidation CuO already form Cu dendrites on the surface, sharply in contrast to the well-maintained morphology of DVL-Cu after 50h CO₂RR (Figs. R4c and d).

This manuscript successfully synthesized dense vertical lamellate Cu nanostructures with excellent catalytic performance and prolonged stability. To explain the morphology stability of the DVL-Cu, COMSOL multiphysics simulations are carried out. Electrode current density and electrolyte current density of DVL-Cu and Cu₂O nanocubes during CO₂RR was acquired by COMSOL multiphysics simulations. Figs. R5a and c represent DVL-Cu while 5b and d represent Cu₂O nanocubes. Cu₂O nanocubes generate enhanced electrode current density with a nearly 5-fold increase. High-magnification images indicate that the electrode current density of regions far away from the substrate is higher than those of closer regions. This result could explain the

slight Cu agglomeration on the surface of nanoplates in DVL-Cu after CO₂RR. Meanwhile, the electrolyte current density (correlated to local electrostatic intensity) in the Cu₂O nanocubes system is higher than that in the DVL-Cu system, especially at the corner of nanocubes. Faster dissolution/redeposition process will occur in those regions with higher electrolyte current density, and increased local electrostatic intensity will lead to easier electro-migration of nanostructures. Simultaneously, the higher electrode current density of Cu₂O nanocubes could accelerate the agglomeration process. These simulation results explain why the Cu₂O nanocubes are easier to agglomerate during CO₂RR, which could be extrapolated to other easily-agglomerated nanostructures. Simultaneously, the moderate electrode current density coupled with the modest local electrostatic intensity ensures the structural stability of DVL-Cu during the CO₂RR. These microscopic-level analyses shed light on the structural stability of DVL-Cu.

Meanwhile, CuO nanoplates are grown in-situ on Cu foil or Cu-evaporated carbon paper. Better charge transfer and binder-free micro-circumstance would assist the catalytic process. The more robust combination with the substrate prevents the migration of Cu nanoplates, which helps avoid the agglomeration of DVL-Cu [Grosse, P. et al. *Angew. Chem. Int. Ed.* 57, 6192-6197 (2018)].

Secondly, the good stability of numerous delicate Cu/Cu₂O interfaces in KCl electrolyte is another key factor for its good performance. The choice of electrolytes is an important issue that can be easily overlooked. Significant differences exist between the catalytic performance obtained from the KHCO₃ system and non-buffering electrolytes, which could be discussed in detail in comment 3 (Fig. R6). More importantly, the high local pH generated by KCl electrolyte can help stabilize the delicate Cu₂O nanoparticles on the surface, where apparent agglomeration of Cu₂O nanoparticles in KHCO₃ is observed (See Figs. R8-10 and comment 3 for the details).

Cl⁻ specific adsorption further assists the suppression of hydrogen evolution in high negative bias during CO₂RR, resulting in higher C₂H₄ faradaic efficiency [Kotaro et al., *Electrochimica Acta*, 56, 2010, 381-386]. (See comment 3 for the details).

In short, the combination of delicate and stable nanostructures, three-dimensional self-supporting structure, and the utilization of non-buffering electrolyte are the most significant differences between this work and previous studies. These data were added in the revised manuscript (Supplementary Note 1) and the corresponding discussion is added in the main text.

Fig. R1 | **a,b**, TEM image of the CuO-NPs at different magnifications. **c**, AFM image of the CuO-NPs and the corresponding height profile from the dashed line. **d**, TEM image of the DVL-Cu.

Fig. R2 | **a,b**, SEM images of Cu₂O nanocubes at different magnifications. **c,d**, SEM images of Cu₂O nanocubes after 1h CO₂RR at different magnifications.

Fig. R3 | **a,b**, SEM images of thermal oxidation CuO at different magnifications. **c,d**, SEM

images of thermal oxidation CuO after 1h CO₂RR at different magnifications.

Fig. R4 | **a,b**, SEM images of CuO-NPs at different magnifications. **c,d**, SEM images of DVL-Cu after 50h CO₂RR at different magnifications.

Fig. R5 | **a,c**, Electrode current density and electrolyte current density distribution of DVL-Cu at 100 mA/cm². **c,d**, Electrode current density and electrolyte current density distribution of Cu₂O

nanocubes at 100 mA/cm^2 . The electrode current density and electrolyte current density (in **a** and **b**) correspond to the left and right legends, respectively. The electrode current density in **c** and **d** correspond to the left and right legends, respectively. The conical arrows represent the electrolyte current density vector.

2. Cu-based catalysts have been widely employed in CO₂-to-C₂H₄ conversion so far, so I would be curious what are the intrinsic differences between this CuO system and previous reports [e.g., Dinh C T et al., Science, 2018, 360(6390): 783-787.] Indeed, the manuscript at present status cannot supply a solid explanation from the microscopic mechanism. In-depth research in this field should be necessary for publishing this work in Nature Communications.

We thank the reviewer for this question. As shown in Fig. R3, CuO systems commonly suffer from agglomeration issues. While some reports focused on improving the catalytic ability of three-phase boundary in the gas-diffusion electrode [e.g., Dinh C T et al., Science, 2018, 360(6390): 783-787.], our research is committed to building delicate nanostructures with stable Cu/Cu₂O interfaces for CO₂-to-C₂H₄ catalysts. The reference is cited in the revised main text and the difference are discussed. As discussed in comments 1 and 3, we believe that the stable nanostructures (Fig. R4) and robust Cu(0)/Cu(I) interfaces (Fig. R10) are the intrinsic differences between this work and previous reports.

3. According to the Abstract in Line 38 and Line 87, the best performance was achieved in neutral KCl electrolyte, which may be more interesting than the performance data. As most of the reported highest performances were obtained in alkaline systems, which typically suffer from a severe carbonate side reaction. A well-established system in neutral electrolyte may be an important progress toward practical application of CO₂RR. But the authors only explained why KCl was chosen at present. The authors should provide a more detailed reason, as well as demonstrate a convincing mechanism on this phenomenon.

We thank the reviewer for pointing out this issue, which helps us understand the mechanism more deeply. To reveal the underlying mechanism of the electrolyte, we selected KHCO₃ (buffer electrolyte) and K₂SO₄ (non-buffer electrolyte without Cl⁻) for analysis. Firstly, the catalytic performance of DVL-Cu in these electrolytes was measured. Total current density plots with

saturated CO₂ are shown in Fig. R6a. Current densities obtained from these electrolytes are very similar. The current density in K₂SO₄ is slightly higher than that in KHCO₃ and KCl. Besides, FEs of products may be more informative. The peak ethylene FE in KHCO₃ and K₂SO₄ is 66.4% (−0.7V) and 72.8% (−0.8V), respectively (Figs. R6b and c). Both ethylene FEs decline when the potential goes further negative, with increased hydrogen FEs. The methane production in K₂SO₄ is nearly negligible. In contrast, methane FE reaches 17.8% in KHCO₃ at −1.1V. By observing the overall product distribution, we can find that the ethylene production in KHCO₃ is not satisfied. The ethylene FEs exceed 60% only in the limited potential range and the methane production is not well suppressed at higher overpotentials. Ethylene FEs in K₂SO₄ at moderate overpotentials are similar to KCl, but hydrogen production dominates at higher overpotentials.

The performance difference can be explained by the following reasons. KHCO₃ is a buffering electrolyte and the local generated OH[−] during CO₂RR could be neutralized by HCO₃[−] leading to lower local pH than the non-buffering electrolyte. We conclude that the lower local pH in KHCO₃ results in inferior ethylene production because lower pH regions could be favorable to hydrogen evolution and methane production, both needing H⁺ [Hori et al., J. Chem. Soc., Faraday Trans. 1, 1989,85, 2309-2326]. Then, higher local pH generated in non-buffering electrolytes would suppress methane production in all potential ranges and hydrogen evolution at lower overpotentials. Comparing the catalytic performance in KCl and K₂SO₄, it could be concluded that the existence of Cl[−] could suppress hydrogen evolution even at high overpotentials, which is considered the result of the Cl[−] specific adsorption effect. The strongly adsorbed Cl[−] could facilitate the electron transfer from the electrode to CO₂ and suppress the adsorption of protons, leading to a higher hydrogen evolution overpotential [Kotaro et al., Electrochimica Acta, 56, 2010, 381-386].

Subsequently, long-term performances in different electrolytes were tested to explore the relationship between catalyst stability and electrolyte (Fig. R6d). Interestingly, the DVL-Cu delivered ~50h stable current density and ethylene FE in K₂SO₄ under −0.8V. In comparison, it only preserved stable performance for less than 10h in KHCO₃ under the same condition. SEM and TEM images of the post-CO₂RR sample in K₂SO₄ (Fig. R7) display equally morphology and Cu(I)/Cu(0) interfaces with that gained from the KCl sample, indicating that high local pH plays a critical role for the prolonged stability. Detailed post-electrolysis characterizations were

performed to unravel the underlying mechanism of stability difference for the sample obtained from KHCO_3 and KCl . As shown in Figs. R8c and d, a more severe Cu agglomeration on the top of nanoplates can be seen from the KHCO_3 sample. AC-TEM and EELS mapping were carried out to distinguish both samples' nanostructure and Cu(I) species distribution. Both samples are nanoplates composed of tiny nanoparticles with almost the same morphology (Figs. R9a, 10a). EELS mapping of the KHCO_3 sample (Figs. R9b-d) reveals that Cu(I) species agglomerate on the top of nanoplates while these species distribute uniformly in the KCl sample (Figs. R10d-f). AES depth profile analyses (Fig. R11) further verify the Cu(I) species distribution differences, where the Cu(I) content in KCl sample is higher along with depth compared to the KHCO_3 sample. Apparently, non-buffering electrolytes could stable the DVL-Cu catalyst by protecting its Cu(I)/Cu(0) interfaces during electrolysis. Since the K_{sp} of CuOH is relatively low (1.0×10^{-14}), high local pH could significantly slow down the dissolution of Cu(I) species. Hence, high local pH would suppress the dissolution/redeposition process of Cu(I) species in non-buffering electrolytes, which preserve the Cu(I)/Cu(0) interfaces during CO_2RR process.

Moreover, based on COMSOL simulation, the top of nanoplates possess a larger surface current density, which leads to higher local pH. It means the bottom region of nanoplates is easier to dissolve. The dissolved Cu(I) species would redeposit more likely on the top of nanoplates with higher local pH, which is consistent with SEM images of the KHCO_3 sample. These data were added in the revised manuscript (Note 1 in the Supplementary information) and the corresponding discussion is added in the main text and Supplementary Note 1.

Fig. R6 | **a**, FEs of the DVL-Cu in 0.5 M KCl. **b**, FEs of the DVL-Cu in 0.25 M K₂SO₄. **c**, FEs of the DVL-Cu in 0.5 M KHCO₃. **d**, LSV curves of DVL-Cu in 0.5 M KCl, 0.5 M KHCO₃ and 0.25 M K₂SO₄ during CO₂RR. **e**, Stability test of DVL-Cu in 0.5 M KCl, 0.5 M KHCO₃ and 0.25 M K₂SO₄ at -0.8V.

Fig. R7 | **a**, SEM image of the 50h post-electrolysis DVL-Cu in K₂SO₄. **b,c**, TEM images of the 50h post-electrolysis DVL-Cu in K₂SO₄ at different magnifications.

Fig. R8 | **a,b**, SEM image of the 50h post-electrolysis DVL-Cu in KCl at different magnifications. **c,d**, SEM images of the 20h post-electrolysis DVL-Cu in KHCO₃ at different magnifications.

Fig. R9 | **a**, STEM bright-field image of the 20h post-electrolysis DVL-Cu in KHCO₃. **b-d**, EELS

maps of Cu, Cu₂O and their overlay in the 20h post-electrolysis DVL-Cu in KHCO₃.

Fig. R10 | **a**, STEM bright-field image of the 50h post-electrolysis DVL-Cu in KCl. **b,c**, EDS maps of Cu, O element in the 50h post-electrolysis DVL-Cu in KCl. **d-f**, EELS maps of Cu, Cu₂O and their overlay in the 50h post-electrolysis DVL-Cu in KCl.

Fig. R11 | **a**, Cu LMM Auger spectra of the 20h post-electrolysis DVL-Cu in KHCO₃ with respect to different Ar⁺ etching depths. **b**, Cu LMM Auger spectra of the 50h post-electrolysis DVL-Cu in KCl with respect to different Ar⁺ etching depths.

4. As indicated in the article's title, the long-term stability is important for this work's novelty. However, the authors didn't give a critical analysis from the performance. For instance, why Cu(0)/Cu(I) interfaces, a thermodynamic factor according to authors' DFT calculations, could result in such a stable activity (i.e., dynamic phenomenon), which has not been reported in previous Cu/Cu₂O studies?

We thank the reviewer for pointing out the question. As revealed in comments 1-3, the long-term stability of DVL-Cu is derived from the following intrinsic factors: 1) Superior structural stability of DVL-Cu. The agglomeration of nanostructures is the main reason for poor stability in most cases. Therefore, superior structural stability is an essential precondition for long-term stability (as shown in Figs. R4b and c). 2) Correct electrolyte selection. As explained in comment 3 (Figs. R6-11), high local pH generated by non-buffering electrolytes effectively slows down the dissolution/redeposition process. Hence, Cu(I)/Cu(0) interfaces are well preserved in these electrolytes. These data were added in the revised manuscript (Fig. 6 in the main text, Supplementary Note 1) and the corresponding discussion is added in the Supplementary Note 1.

5. To follow up the previous question, what were the catalyst structures after the 55 hours of flow cell tests? Could these Cu(0)/Cu(I) interfaces be maintained after this long-term test at 150 mA cm⁻²? The authors should provide detailed post-electrolysis characterizations and give a clear explanation. Besides, EIS cannot supply much useful information in such a multistep reaction with unknown pathways.

We thank the reviewer very much for mentioning this important issue. We agree with the reviewer that EIS cannot supply much useful information in such a multistep reaction with unknown pathways. As requested by the reviewer, detailed post-electrolysis characterizations (SEM, TEM, AC-TEM, EELS mapping and AES depth profile) in KCl electrolyte coupled with failure analysis in KHCO₃ electrolyte were performed to unravel the underlying mechanism of stability. The nanoplates densely stack in order without any agglomeration after 50h electrolysis, only a slight increase of plates thickness is observed (Figs. R8a and b). AC-TEM images display that the nanoplates are composed of tiny nanoparticles, consistent with the 2h post-electrolysis sample. Elemental mapping shows the uniform distribution of Cu and O elements in the nanoplates (Figs. R10b and c). Valence state analyses exhibit numerous Cu(I)/Cu(0) interfaces without any

agglomeration (Figs. R10d-f). AES depth profiles (Fig. R11b) reveal that subsurface Cu(I) content is substantial, which is sharply contrasting with that in KHCO_3 . We argue that the moderate surface current density and local electrostatic intensity coupled with the stable self-standing nanostructures guarantee superior structural stability of DVL-Cu. In addition, high local pH generated by non-buffering electrolytes effectively slows down the dissolution/redeposition process, Cu(I)/Cu(0) interfaces are well preserved in these electrolytes. In brief, stable nanostructure and steady Cu(I)/Cu(0) interfaces result in the prolonged stability of DVL-Cu. These data were added in the revised manuscript (Note 1 in the Supplementary information) and the corresponding discussion is added in the main text.

6. In the Supporting information, the authors presented the 3D optical profiler simulated images and calculated results of CuO; and in Line 128, the authors stressed that “The Ra of CuO-NPs is 128 5.96 μm , dramatically larger than that of CuO (3.730 μm)”. Later in the text (Line 223), roughness factor (RF) was also calculated by double-layer capacitance method. So, what is the difference between ECSA and roughness factor? And what can these data be useful for the CO₂RR evaluation?

We thank the reviewer very much for mentioning this interesting issue. The Ra measured by 3D optical profiler is different from the roughness factor measured by the double-layer capacitance method. The former indicates physics roughness while the latter is equal to ECSA. ECSA is defined as the ratio of the reactive area of catalyst to that of pure copper. RF is obtained by dividing the double-layer capacitance (equal to the reactive area) of the catalyst by the double-layer capacitance of ideal smooth copper, which is a standard method to calculate ECSA [Sachin et al., *Advanced Powder Technology* 29 (2018) 3520–3526]. Therefore, RF is the result of calculated ECSA. For CO₂RR catalysts, delicate nanostructures could increase the number of electrochemically active sites, increasing ECSA value and improving reaction activity. Hence, ECSA can be used as a reference standard for the reaction activity of catalysts. On the other hand, physics roughness (Ra) can be used as an indicator of ECSA in most cases, and usually a higher physics roughness (Ra) means increased ECSA. Thus, we further used the Ra to illustrate the exquisite nanostructures of the designed catalysts.

Reviewer #2 (Remarks to the Author):

This paper presents a combined experimental and DFT study of CO₂ electrochemical conversion to C₂ products on CuO/Cu catalysts. The authors need to address the following points before it is considered as a potential Nat. Commun publication:

(i) The authors claimed that C₂H₄ is the main product. However, Figure 4e clearly indicates that ethanol is the main product. The same figure also shows that formate is one of the main products. There is no C₂H₄ in Figure 4c.

We thank the reviewer for pointing out the question. The authors would like to display ratios of liquid products in Figure 4e. However, this form was inappropriate due to the lack of Faradic efficiencies and the misleading expression. We replaced the “ratios” in the Y-axis of Figure 4e to “FEs”, which clearly shows the precise liquid FEs of catalysts at different potentials and effectively avoids misunderstanding (Fig. R12). These data were updated in the revised manuscript (Fig. 4e in the main text) and the corresponding discussion is added in the main text.

Fig. R12 | Liquid products FEs of DVL-Cu and R-CuO-NPs.

(ii) As the authors mentioned HER is always a competing reaction. The HER free energy diagram is not included and discussed in the manuscript. The authors should perform DFT to get HER energetics and should be compared with the CO₂RR free energy diagrams to show that HER is less favorable and the catalysts selectively promote the CO₂RR.

We thank the reviewer for pointing out this question. We have calculated HER energetics of three

catalytic sites and compared them with CO₂RR free energy diagrams (Fig. R13). DFT results present that the HER barriers of three catalytic sites (0.82 eV on Cu(110) slab, 1.44 eV on Cu/Cu₂O interface and 0.72 eV on Cu₂O(110) slab) are higher than the ethylene rds (rate-determined step) free energy barriers of Cu/Cu₂O site (0.60 eV). Therefore, it can be proved that CO₂RR is more favorable than HER in the DVL-Cu system. Also, the HER barrier of Cu/Cu₂O interface is the highest, consistent with the experimental results that H₂ FE of DVL-Cu is the lowest. These data were updated in the revised manuscript (Fig. 54 in the Supplementary information) and the corresponding discussion is added in the main text.

Fig. R13 | A reaction energy diagram for HER on Cu(110) slab, Cu/Cu₂O interface and Cu₂O(110) slab.

(iii) As shown in Figure 4c, the authors should also include and discuss the formate pathway and compared with the HOCO pathway which is claimed to produce C₂ products.

We thank the reviewer very much for mentioning this important issue. We have calculated different paths of the formate formation process on three catalytic sites and compared them with free energy barriers of C₂ pathway (Fig. R14), such as *CO₂+H→*HCOO (ΔG = 0.80 eV on the Cu/Cu₂O interface), *CO₂+H→*COOH (ΔG = 0.50 eV on the Cu/Cu₂O interface), *COOH+H→*HCOOH (ΔG = 1.57 eV on the Cu/Cu₂O interface) and *COOH+H→*CO (ΔG = -0.32 eV on the Cu/Cu₂O interface). By comparing their free energy barriers, it indicates that the energy barriers of formate formation (through *HCOO or *COOH intermediate) are higher than

that of C2 product pathway. Finally, it is found that the pathway on the Cu/Cu₂O interface that leads to the *OCCOH formation (C2 product intermediate) is the optimal one. Therefore, it can be proved that C2 products are more favorable than formate in the DVL-Cu system. These data were updated in the revised manuscript (Fig. 53 in the Supplementary information) and the corresponding discussion is added in the main text.

Fig. R14 | A reaction energy diagram for CO₂RR on Cu(110) slab, Cu/Cu₂O interface and Cu₂O(110) slab. Intermediates corresponding to lighter colors are marked in red font.

(iv) Since ethanol is one of the main C2 products, it would be helpful to have free energy diagram that leads to the formation of C₂H₅OH and be compared with the C₂H₄ pathway.

We thank the reviewer very much for mentioning this important issue. We have calculated the catalytic path of ethanol (*OHC₂H₂→*OHC₂H₃→*OHC₂H₄→*+OHC₂H₅) on three catalytic sites (Fig. R15). After comparison, the first divergent step (*OHC₂H₂→*OHC₂H₃, ΔG = 2.32 eV on Cu/Cu₂O interface) of ethanol is unfavorable to proceed compared with ethylene pathway (*OHC₂H₂→*O+C₂H₄, ΔG = -0.52 eV on Cu/Cu₂O interface) on three catalytic sites. Therefore, DFT results match well with the low FE of ethanol (< 7%) in the DVL-Cu system. These data were updated in the revised manuscript (Fig. 55 in the Supplementary information) and the corresponding discussion is added in the main text.

Fig. R15 | A reaction energy diagram for CO₂RR to ethylene and ethanol on Cu(110) slab, Cu/Cu₂O interface and Cu₂O(110) slab. Intermediates corresponding to lighter colors are marked in red font.

(v) The authors assumed that CH₄ forms from COH intermediate. However there are several other channel that lead to the formation of CH₄. For example: $\text{CO}_2 \rightleftharpoons \text{HCOO} \rightleftharpoons \text{HCOOH} \rightleftharpoons \text{H}_2\text{COOH} \rightleftharpoons \text{H}_2\text{CO} \rightleftharpoons \text{H}_3\text{CO} \rightleftharpoons \text{CH}_3 \rightleftharpoons \text{CH}_4$. There is no rational presented why other channels are not included in the discussion.

We thank the reviewer for pointing out the question. We do not explain this clearly in the first version. CO₂RR is a continuous proton-coupled electron transfer (PCET) process. In a single hydrogenation step, there are several reaction sites, which will lead to different products ultimately. Free energy barriers of different hydrogenation steps that lead to methane and C₂ products were calculated on three catalytic sites. As shown in Fig. R14, the first PCET step that form *COOH intermediate requires lower free energy than that form *HCOO intermediate, which indicates the PCET process tends to occur through *COOH intermediate on three catalytic sites. Based on the optimal intermediate of the previous step, the free energy barriers of *CO and *HCOOH intermediate generated from *COOH have been calculated to distinguish the reaction trend. Combining the DFT results, the CO₂RR tends to process through: $\text{CO}_2 \rightleftharpoons \text{*COOH} \rightleftharpoons \text{*CO} \rightleftharpoons \text{*CO+CO} \rightleftharpoons \text{*CO+COH} \rightleftharpoons \text{*OCCOH}$ (C₂ pathway) on three

catalytic sites. Meanwhile, the most possible CH₄ pathway (if proceed) is CO₂==>*COOH==>*CO==>*COH. However, insurmountable energy barriers are required for the hydrogenation of *CO to form *COH (*CO+H→*COH, reaction step of CH₄ formation pathway) on three catalytic sites, which leads the CO₂RR to C₂ products instead of CH₄. These data were updated in the revised manuscript (Fig. 53 in the Supplementary information) and the corresponding discussion is added in the main text.

(vi) The selection of DFT model for the interface needs to be justified. Did authors test other models (smaller/larger nanoparticle)? What is the size of Cu₂O nanoparticle in experiment? Does the DFT model correspond to experiments?

We thank the reviewer very much for pointing out this important issue. We have validated the computational model when constructing the model. Firstly, we calculated the adsorption area ratio of Cu₂O nanoparticles (~48%), Cu/Cu₂O (~6.18:1) ratio and the size of Cu₂O nanoparticles (4-12 nm) from the experimental results. Different expansion ratios of Cu(110) surface, including 3*3, 4*4, 5*5, 3*6, were simulated to verify their stability and structural property. Subsequently, it is found that to achieve the surface adsorption ratio of Cu₂O nanoparticles (50%) and Cu/Cu₂O ratio (6:1), the expansion ratio of 3*6 (16.58 Å*8.31 Å*25.84 Å) is the most appropriate. Scaling down the model equally would lead to structural instability of Cu/Cu₂O interfaces while scaling up the model would exceed the state-of-the-art computational capability. In fact, the 3*6 model possesses all the structural characteristics and identical interface properties with the DVL-Cu system. Therefore, the DFT model for the interface is consistent with experimental results and could demonstrate catalytic performance of the DVL-Cu catalyst.

Reviewer #3 (Remarks to the Author):

Wei Liu et al., presented a facile synthesis method to prepare Cu-based electrocatalyst for the CO₂RR. They have shown that their catalyst, called dense vertical lamellate Cu nanoparticles (DVL-Cu), under the reduction applied potential maintains the Cu₂O/Cu interface to keep ethylene Faradaic efficiency (FE) at high value for a long time, while with a considerably improved total energy efficiency (EE). The attained FE for ethylene is 84.5%, and stable for 55

hours, which is a significant improvement compared to the previous reports so far. A thorough set of characterizations is performed to shed light on the relation between catalyst composition/structure and its superior activity. Overall, this reviewer is confident to recommend the manuscript to be published in the journal Nature Communications, after considering the following suggestions.

In page 12, line 252, it is recommended to add a SEM image of DVL-Cu after 2h CO₂RR (and through comparison) showing that the agglomeration of Cu nanoparticles is the main reason for the instability.

We thank the reviewer very much for pointing out this important issue. We compared the SEM images of DVL-Cu and R-CuO-NPs taken after 2h CO₂RR. As shown in Figures R16c and d, the agglomeration of the nanoplates to nanoparticles is observed after 2h CO₂RR of R-CuO-NPs. In contrast, the SEM images of DVL-Cu (Figures R16a and b) indicate that the vertically arranged and densely stacked laminated nanostructures are retained after 2h electrochemical reduction, which shows the agglomeration of Cu nanoparticles is the main reason for the instability. These data were added in the revised manuscript (Fig. 22 in the Supplementary information) and the corresponding discussion is added in the main text.

Fig. R16 | **a,b**, SEM image of the 2h post-electrolysis DVL-Cu at different magnifications. **c,d**, SEM image of the 2h post-electrolysis R-CuO-NPs at different magnifications.

The author emphasized the capability of this method for industrialization, yet using KCl as the electrolyte is quite hazardous, and it might be a big challenge for industrialization.

We thank the reviewer for pointing out the question. We agree with the reviewer that chloride ion (Cl^-) could discharge at the anode in KCl electrolyte, harmful to the device and environment. We did observe that the Cl^- in catholyte would be transported through the membrane to anolyte when anion exchange membrane (AEM) was used in the flow cell system. Fortunately, although KCl as catholyte was used, the anolyte choice of H-cell system was arbitrary without difference in catalytic performance. Meanwhile, the Nafion-117 membrane can be used as the membrane, which can completely block Cl^- transmission. As shown in Figure R17, the Nafion membrane system display nearly the same FEs of ethylene and full-cell EEs at different current densities with the AEM system (KHCO_3 was chosen as the anolyte). Therefore, the catalytic performance of DVL-Cu was irrelevant with the anolyte and ion exchange membrane. Those anolytes with no Cl^- and membranes that could block the transportation of Cl^- would be suitable for industrialization in the future. These data were added in the revised manuscript (Fig. 34 in the Supplementary information) and the corresponding discussion is added in the Supplementary information.

Fig. R17 | Ethylene FE and EE of DVL-Cu in flow cell using Nafion membrane.

In Fig. 4e, it is recommended to plot FE rather than ratios, albeit the FEs are quite small. Otherwise, this reviewer does not find the figure informative. Also, please use the same potential range for Fig. 4 A and B.

We thank the reviewer for pointing out this. The liquid FEs of DVL-Cu and R-CuO-NPs at different potentials are plotted in Figure R18. Meanwhile, the same potential ranges from -0.6 V to -1.1 V were chosen in Figs. 4A and B to compare their catalytic performance. Now, this figure is more informative (Fig. R19). These data were updated in the revised manuscript (Figs. 4a and 4e in the main text) and the corresponding discussion is added in the main text.

Fig. R18 | FEs of liquid products of DVL-Cu and R-CuO-NPs.

Fig. R19 | FEs of DVL-Cu.

In line 230 and 231, the author mentioned the generation of H₂ is severely suppressed. 21% FE is not considered a “severely” suppression. Also, total FEs are not reported. From the current results it seems the total FE goes above 100%. In addition, the rationale provided in line 232 for the H₂ suppression is not satisfying. This reviewer suggests testing both catalysts for the HER and compare the results. It is possible that the DVL-Cu intrinsically is less active for the HER.

We thank the reviewer very much to mention this important issue. HER tests were performed on both catalysts, and results were plotted with LSV curves obtained during CO₂RR. As shown in Fig. R20, the current density of DVL-Cu during CO₂RR is much higher than that in the HER condition, indicating intrinsic higher CO₂RR activity than HER. Current densities of R-CuO-NPs have a similar trend, but the enhancement is much smaller than that of DVL-Cu. From Fig. R20 we can conclude that the two catalysts have similar intrinsic HER activity, while the DVL-Cu has much higher intrinsic CO₂RR activity than R-CuO-NPs. Thus, the high ethylene FE of DVL-Cu is originated from the higher intrinsic CO₂RR activity rather than the suppression of HER. We changed the description of “severely suppression of H₂” to “the FE_{H₂} is only 13.6% at -0.9 V of DVL-Cu”.

In addition, we carefully checked FEs and found that only the total FEs at -0.9V of DVL-Cu was slightly beyond 100%. This phenomenon violates scientific principles but was common in the CO₂RR research field (for example, in *Science* 372, 1074 2021). The errors usually arise from the quantification of gaseous and liquid products. We retested the potential whose total FE went above 100% and list the new total FEs in Table 1. These data were updated in the revised manuscript (Fig. 4a in the main text, Fig. 18 and Table 1 in the Supplementary information) and the corresponding discussion is added in the main text.

Fig. R20 | LSV curves of DVL-Cu and R-CuO-NPs in Ar and CO₂ saturated KCl electrolyte.

Table R1 | FEs of DVL-Cu at different potentials in H-cell.

Potential (V)	FE (%)									
	C ₂ H ₄	H ₂	CO	CH ₄	formate	ethonal	acetate	Pr-OH	C ₂ H ₆	Total
-0.6	43.8	39.7	10.8	0.0	1.6	1.5	0.1	0.5	0.0	97.9
-0.7	63.9	28.1	1.4	0.7	0.6	3.0	0.1	0.4	0.0	98.1
-0.8	69.8	20.4	2.6	0.3	0.9	2.8	0.1	0.8	0.8	98.5
-0.9	74.9	13.6	2.9	0.8	1.0	3.3	0.1	0.8	1.4	98.9
-1.0	66.3	17.2	2.6	4.8	0.7	4.8	0.1	0.8	1.9	99.2
-1.1	60.8	22.3	1.5	5.9	1.4	5.5	0.0	0.4	0.5	98.3

In lines 248-252: why two catalysts are tested at two different potentials for the stability test? They must be compared under similar conditions, otherwise, drawing a conclusion would not be possible. Either test either DVL-Cu at -1.0V or R-CuO-NPs at -0.8V.

We thank the reviewer for pointing out the questions. It is our fault for testing the two catalysts at different potentials for the stability test. We tested R-CuO-NPs at -0.8V and depicted the results in Fig. R21. As shown in Fig. R21, the DVL-Cu delivers a steady i-t curve and high C₂H₄ selectivity for 50h at -0.8V. In contrast, the R-CuO-NPs only retains stable FEC₂H₄ for less than 7h. These data were updated in the revised manuscript (Fig. 21 in the Supplementary information) and the corresponding discussion is added in the main text.

Fig. R21 | Stability test of DVL-Cu and R-CuO-NPs at constantly applied potentials.

Lines 253-261: although Cu₂O was shown in the HRTEM for DVL-Cu, it is recommended adding GI-XRD profiles after 1h CO₂RR, like what has been done for the R-CuO-NPs.

As requested by the reviewer, GI-XRD profiles after 1h and 2h CO₂RR were provided in Fig. R22, and Fig. 3a in the main text, respectively. As shown in these images, the Cu₂O characteristic peak remains after 1h or 2h CO₂RR, which is absent in the R-CuO-NPs profiles after 1h CO₂RR. Meanwhile, the peak intensity of R-CuO-NPs is higher than that of DVL-Cu, indicating that it has a larger crystal size. This could correspond to Cu nanoparticles' agglomeration of R-CuO-NPs, as shown in Fig. R16. These data were updated in the revised manuscript (Fig. 24 in the Supplementary information) and the corresponding discussion is added in the Supplementary information.

Fig. R22 | GI-XRD profile of the R-CuO-NPs and DVL-Cu taken after 1 h CO₂RR.

In the table 1, the partial current density of C₂H₄ was mentioned 174.4 mA/cm² which does not seem to be super accurate. At -0.81V the partial current density was mentioned 92.5 mA/cm² (line

289).

We thank the reviewer for this question. The current density of C₂H₄ does have some mistakes because it reaches 174.4 mA/cm² at -1.01V instead of -0.81V. The author's original purpose was to show the maximum current density of C₂H₄ but these means of expression were easy to be misunderstood. Now we update Table R2, in which the potentials, FEs and current densities correspond to each other. These data were updated in the revised manuscript (Table 1 in the main text) and the corresponding discussion is added in the main text.

Catalysts	Electrolyte	Potential (V vs. RHE)	FE _{C₂H₄} (%)	j _{C₂H₄} (mA cm ⁻²)	Maximum EE _{C₂H₄} (%)*	Reference
DVL-Cu	0.5 M KCl	-0.81	84.5	92.5	28	This work
		-1.01	67.1	174.4		
OBC	0.5 M KHCO ₃	-1.00	45	44.7	/	20
O ₂ -plasma-treated Cu	0.1 M KHCO ₃	-0.9	60	7.2	/	21
Cu 3D CTPI	/	/	69	304	22	51
Cu(B)-2	0.1 M KCl	-1.1	52	36.4	/	14
Cu-12	1 M KHCO ₃	-0.83	72	230	20	52

*The maximum EE_{C₂H₄} was not achieved in the potentials listed in the table.

Line 299, the author claimed this is the highest value achieved in the "neutral" catholyte. This requires further evidence, e.g., measuring pH, during and (especially after the long stability experiment). A plot of pH vs. time will be very informative.

We thank the reviewer very much to mention this important issue, which makes our work more solid. As requested by the reviewer, pH value of the catholyte was measured during long-term stability experiment. In the H-cell system (Fig. R23), pH value raised rapidly from 3.90 to 6.52 in the initial 1h. Then pH value retained around 6.7 for the following test, a neutral catholyte environment. Meanwhile, the FE of C₂H₄ remained stable in this pH range, indicating good catalytic performance could be achieved in the neutral catholyte.

Moreover, during the long-term stability experiment of the flow cell system, the catholyte was replaced every 6h in the previous test. To monitor the variation of pH value of catholyte during the test, the catholyte remained unchanged for 24h. As shown in Fig. R24, pH value raised from 6.95 to 8.23 in the first 1h, and reached 9.50 after 5h electrolysis. The pH value fluctuated around 9.8 for the rest of time. Despite pH changing over time, the FE of C₂H₄ remained stable around 80%, demonstrating that DVL-Cu delivers similar catalytic performance in the pH range from 6.5 to 9.8. In industrial production, the pH of the solution can be easily adjusted near neutral, which could minimize corrosion to equipment without affecting the yield. These data were updated in the revised manuscript (Figs. 19 and 42 in the Supplementary information) and the corresponding discussion is added in the Supplementary information.

Fig. R23 | pH and ethylene FE during the long-term test in H-cell.

Fig. R24 | pH and ethylene FE during the long-term test in flow cell.

The manuscript needs another round of proofreading as there are several typos and sentences with grammatical errors which make it difficult to understand. Few examples are mentioned below:

- Line 32: “an facile”
- Fig. 1c: CuO growth and CuO-nucleation graph caption should be revised.
- Supplementary Fig. 10 caption: R-CuO instead of A-CuO. This is repeated many times.
- Line 93: “catholic” reduction
- Supplementary Fig. 20 caption is incorrect.

We thank the reviewer very much. The above mistakes have been completely corrected now.

- Line 150: the authors discuss the decrease in Cu (110) peak intensity but no peak is assigned to this facet in Fig. 2d.

Copper belongs to the face-centered cubic system. Cu (110) peak would be eliminated in X-ray diffraction. Cu (220) facet is the equivalent crystal plane of Cu (110), which has the same relationship as Cu (200) and Cu (100) in Fig. 2d. Therefore, we could judge the peak intensity variation trend of Cu (110) from that of Cu (220) in Fig. 2d.

- Fig. 2F: don’t we expect to observe an intensified Cu₂O peak at -1V compared to that at -0.6V after 1800 s?

Cu₂O would be gradually reduced to Cu⁰ under negative potentials according to electrochemical principles. Either extending the reducing time or improving the negative bias will lead to the

decrease of the Cu₂O ratio in the catalyst, which could also be seen in previous studies like **Nature Catalysis** volume 1, pages 103–110 (2018). Therefore, it is reasonable that weaker Cu₂O was seen at $-1V$ compared to $-0.6V$.

REVIEWERS' COMMENTS

Reviewer #1 (Remarks to the Author):

This revised manuscript at current stage shows good improvement on the study of CuO-based CO₂RR and reveals the atomic-scale mechanism (i.e., Cl⁻ anion-induced stable Cu(0)/Cu(I) interface as well as higher local pH) for efficient CO₂-to-C₂H₄ conversion. As a result, from the perspective of internal mechanism, research novelty, and methodology, the revised manuscript can be accepted in Nature communication with the following minor revision.

Suggestion: Based on the response by the authors, “physics roughness (Ra) can be used as an indicator of ECSA in most cases, and usually a higher physics roughness (Ra) means increased ECSA”, we would conclude that physics roughness (Ra) plays almost the same role with ECSA. According to Occam's Razor, the concept of Ra is not necessary to be proposed in this work when ECSA can be obtained. We'd suggest authors to delete the Ra-related description and rewrite this part to make this work more concise. Otherwise, readers outside this field may be confused.

Reviewer #2 (Remarks to the Author):

The authors have addressed most of the questions. However, the line of argument presented for the DFT model construction of Cu₂O/Cu is not convincing. The binding strength of intermediates depend on the side of the Cu₂O cluster on Cu surface. The authors should test few different clusters of different size and calculate the CO₂RR free energy diagrams to show that the DFT conclusions remain same and agree with the experimental trend.

Reviewer #3 (Remarks to the Author):

none

REVIEWER' COMMENTS

Reviewer #1 (Remarks to the Author):

This manuscript employed an oxygen-rich ultrathin CuO nanoplates arrays, in which Cu/Cu₂O heterogeneous interfaces were formed, for electrochemical CO₂ reduction reaction (CO₂RR) process. As presented in the manuscript, the performance was good (i.e., 84.5% selectivity toward C₂H₄ at -0.8 V with a current density of ~100 mA cm⁻²), and the Cu(0)/Cu(I) interfaces related structure-activity relationship was proposed. It was suggested by authors that the insurmountable energy barrier for the hydrogenation of single adsorbed *CO and the facile C₂H₄ formation pathway on the Cu(110)/Cu₂O(110) was the possible reason accounting for superior C₂H₄ selectivity. However, from the perspective of internal mechanism, research novelty, and methodology, this work still does not meet the similar level with recent alike publications in Nature Communications. Detailed comments are given as following:

1. The authors reported that oxygen-rich ultrathin CuO nanoplates arrays delivered a high selectivity toward C₂H₄ in CO₂RR, and the authors concluded the mechanism to be a Cu(0)/Cu(I) interfaces effect. However, similar researches on Cu/Cu₂O or Cu(0)/Cu(I) have been extensively reported [e.g., Altaf N et al., Journal of Energy Chemistry, 2020, 48: 169-180; Yan Z et al. Catalysis Science & Technology, 2021]. Therefore, it is not enough to supply a simple DFT calculation, as well as a series of experiments for confirming the formation of copper/cuprous oxide interfaces, for supporting the authors' viewpoint. The authors should clearly describe the differences between this work and previous studies, instead of simply listing the comparison of the CO₂RR performances.
2. Cu-based catalysts have been widely employed in CO₂-to-C₂H₄ conversion so far, so I would be curious what are the intrinsic differences between this CuO system and previous reports [e.g., Dinh C T et al., Science, 2018, 360(6390): 783-787.] Indeed, the manuscript at present status cannot supply a solid explanation from the microscopic mechanism. In-depth research in this field should be necessary for publishing this work in Nature Communications.
3. According to the Abstract in Line 38 and Line 87, the best performance was achieved in neutral KCl electrolyte, which may be more interesting than the performance data. As most of the reported highest performances were obtained in alkaline systems, which typically suffer from a severe carbonate side reaction. A well-established system in neutral electrolyte may be an important progress toward practical application of CO₂RR. But the authors only explained why KCl was chosen at present. The authors should provide a more detailed reason, as well as demonstrate a convincing mechanism on this phenomenon.
4. As indicated in the article's title, the long-term stability is important for this work's novelty. However, the authors didn't give a critical analysis from the performance. For instance, why Cu(0)/Cu(I) interfaces, a thermodynamic factor according to authors' DFT calculations, could result in such a stable activity (i.e., dynamic phenomenon), which has not been reported in previous Cu/Cu₂O studies?

5. To follow up the previous question, what were the catalyst structures after the 55 hours of flow cell tests? Could these Cu(0)/Cu(I) interfaces be maintained after this long-term test at 150 mA cm^{-2} ? The authors should provide detailed post-electrolysis characterizations and give a clear explanation. Besides, EIS cannot supply much useful information in such a multistep reaction with unknown pathways.

6. In the Supporting information, the authors presented the 3D optical profiler simulated images and calculated results of CuO; and in Line 128, the authors stressed that "The Ra of CuO-NPs is $128.596 \mu\text{m}$, dramatically larger than that of CuO ($3.730 \mu\text{m}$)". Later in the text (Line 223), roughness factor (RF) was also calculated by double-layer capacitance method. So, what is the difference between ECSA and roughness factor? And what can these data be useful for the CO₂RR evaluation?

Reviewer #2 (Remarks to the Author):

This paper presents a combined experimental and DFT study of CO₂ electrochemical conversion to C₂ products on CuO/Cu catalysts. The authors need to address the following points before it is considered as a potential Nat. Commun publication:

(i) The authors claimed that C₂H₄ is the main product. However, Figure 4e clearly indicates that ethanol is the main product. The same figure also shows that formate is one of the main products. There is no C₂H₄ in Figure 4c.

(ii) As the authors mentioned HER is always a competing reactions. The HER free energy diagram is not included and discussed in the manuscript. The authors should perform DFT to get HER energetics and should be compared with the CO₂RR free energy diagrams to show that HER is less favorable and the catalysts selectively promote the CO₂RR.

(iii) As shown in Figure 4c, the authors should also include and discuss the formate pathway and compared with the HOCO pathway which is claimed to produce C₂ products.

(iv) Since ethanol is one of the main C₂ products, it would be helpful to have free energy diagram that leads to the formation of C₂H₅OH and be compared with the C₂H₄ pathway.

(v) The authors assumed that CH₄ forms from COH intermediate. However there are several other channel that lead to the formation of CH₄. For example: CO₂ ==> HCOO ==> HCOOH ==> H₂COOH ==> H₂CO ==> H₃CO ==> CH₃ ==> CH₄. There is no rational presented why other channels are not included in the discussion.

(vi) The selection of DFT model for the interface needs to be justified. Did authors test other models (smaller/larger nanoparticle)? What is the size of Cu₂O nanoparticle in experiment? Does the DFT model correspond to experiments?

Reviewer #3 (Remarks to the Author):

Wei Liu et al., presented a facile synthesis method to prepare Cu-based electrocatalyst for the CO₂RR. They have shown that their catalyst, called dense vertical lamellate Cu nanoparticles (DVL-Cu), under the reduction applied potential maintains the Cu₂O/Cu interface to keep ethylene Faradaic efficiency (FE) at high value for a long time, while with a considerably improved total energy efficiency (EE). The attained FE for ethylene is 84.5%, and stable for 55 hours, which is a significant improvement compared to the previous reports so far. A thorough set of characterizations is performed to shed light on the relation between catalyst composition/structure and its superior activity. Overall, this reviewer is confident to recommend the manuscript to be published in the journal Nature Communications, after considering the following suggestions.

In page 12, line 252, it is recommended to add a SEM image of DVL-Cu after 2h CO₂RR (and through comparison) showing that the agglomeration of Cu nanoparticles is the main reason for the instability.

The author emphasized the capability of this method for industrialization, yet using KCl as the electrolyte is quite hazardous, and it might be a big challenge for industrialization.

In Fig. 4e, it is recommended to plot FE rather than ratios, albeit the FEs are quite small. Otherwise, this reviewer does not find the figure informative. Also, please use the same potential range for Fig. 4A and B.

In line 230 and 231, the author mentioned the generation of H₂ is severely suppressed. 21% FE is not considered a "severely" suppression. Also, total FEs are not reported. From the current results it seems the total FE goes above 100%. In addition, the rationale provided in line 232 for the H₂ suppression is not satisfying. This reviewer suggests testing both catalysts for the HER and compare the results. It is possible that the DVL-Cu intrinsically is less active for the HER.

In lines 248-252: why two catalysts are tested at two different potentials for the stability test? They must be compared under similar conditions, otherwise, drawing a conclusion would not be possible. Either test either DVL-Cu at -1.0V or R-CuO-NPs at -0.8V.

Lines 253-261: although Cu₂O was shown in the HRTEM for DVL-Cu, it is recommended adding GI-XRD profiles after 1h CO₂RR, like what has been done for the R-CuO-NPs.

In the table 1, the partial current density of C₂H₄ was mentioned 174.4 mA/cm² which does not seem to be super accurate. At -0.81V the partial current density was mentioned 92.5 mA/cm² (line 289).

Line 299, the author claimed this is the highest value achieved in the "neutral" catholyte. This requires further evidence, e.g., measuring pH, during and (especially after the long stability experiment). A plot of pH vs. time will be very informative.

The manuscript needs another round of proofreading as there are several typos and sentences with grammatical errors which make it difficult to understand. Few examples are mentioned below:

- Line 32: "an facile"
- Fig. 1c: CuO growth and CuO-nucleation graph caption should be revised.
- Supplementary Fig. 10 caption: R-CuO instead of A-CuO. This is repeated many times.
- Line 93: "catholic" reduction
- Supplementary Fig. 20 caption is incorrect.
- Line 150: the authors discuss the decrease in Cu (110) peak intensity but no peak is assigned to this facet in Fig. 2d.
- Fig. 2F: don't we expect to observe an intensified Cu₂O peak at -1V compared to that at -0.6V after 1800 s?

Reply to reviewers

Reviewer #1 (Remarks to the Author):

This manuscript employed an oxygen-rich ultrathin CuO nanoplates arrays, in which Cu/Cu₂O heterogeneous interfaces were formed, for electrochemical CO₂ reduction reaction (CO₂RR) process. As presented in the manuscript, the performance was good (i.e., 84.5% selectivity toward C₂H₄ at -0.8 V with a current density of ~ 100 mA cm⁻²), and the Cu(0)/Cu(I) interfaces related structure-activity relationship was proposed. It was suggested by authors that the insurmountable energy barrier for the hydrogenation of single adsorbed *CO and the facile C₂H₄ formation pathway on the Cu(110)/Cu₂O(110) was the possible reason accounting for superior C₂H₄ selectivity. However, from the perspective of internal mechanism, research novelty, and methodology, this work still does not meet the similar level with recent alike publications in Nature Communications. Detailed comments are given as following:

1. The authors reported that oxygen-rich ultrathin CuO nanoplates arrays delivered a high selectivity toward C₂H₄ in CO₂RR, and the authors concluded the mechanism to be a Cu(0)/Cu(I) interfaces effect. However, similar researches on Cu/Cu₂O or Cu(0)/Cu(I) have been extensively reported [e.g., Altaf N et al., Journal of Energy Chemistry, 2020, 48: 169-180; Yan Z et al. Catalysis Science & Technology, 2021]. Therefore, it is not enough to supply a simple DFT calculation, as well as a series of experiments for confirming the formation of copper/cuprous oxide interfaces, for supporting the authors' viewpoint. The authors should clearly describe the differences between this work and previous studies, instead of simply listing the comparison of the CO₂RR performances.

We thank the reviewer for this constructive suggestion. To summarize, the main differences between our work and previous studies are the stability of exquisite nanostructures derived from the ultrathin CuO nanoplates arrays, the stability of numerous delicate Cu/Cu₂O interfaces due to utilization of neutral electrolyte and the assistance of chloride ions. Although much attention has been devoted to the Cu(0)/Cu(I) systems for CO₂RR, the unsatisfactory faradaic efficiency and poor stability still exist in these catalysts.

Firstly, those systems developed in references without exquisite nanostructures usually suffer from severe morphology collapse after being reduced, which diminishes the number of active sites and increases the risk of agglomeration during CO₂RR. Agglomeration of catalysts was familiar in previous research [e.g., Altaf N et al., *Journal of Energy Chemistry*, 2020, 48: 169-180; Yan Z et al. *Catalysis Science & Technology*, 2021] and was a principal cause of poor stability. In the two articles mentioned by the reviewer, there may be problems with vague nanostructures. For example, the flower-shaped Cu₅Al-CO₃ precursor severely agglomerated after being reduced to CO₂RR catalyst in **Journal of Energy Chemistry**, 2020, 48: 169-180. In the revised version, we cited these two references and compared them with our results.

To better answer the question raised by the reviewer, we also compared the morphology evolution of CuO ultrathin nanoplates with other samples. As shown in Figs. R1a-c, CuO nanoplates are only a few atomic layers thick, which have numerous defects and undercoordinated sites. The ultrathin nanoplates can easily self-evolved into well-maintained morphology with numerous active sites exposed (Fig. R1d). The morphology and Cu(I)/Cu(0) distribution were carefully investigated after the long-term test, and details are shown in comment 3 (Figs. R8, 10 and 11). The well-maintained delicate morphology during precursor reduction and CO₂RR process guarantees superior catalytic performance and stability of DVL-Cu. The morphologies of Cu₂O nanocubes and thermal oxidation derived CuO before and after CO₂RR were characterized to display the agglomeration of these unstable nanostructures intuitively. As shown in Figs. R2 and R3, Cu₂O nanocubes severely agglomerate after 1h CO₂RR while the thermal oxidation CuO already form Cu dendrites on the surface, sharply in contrast to the well-maintained morphology of DVL-Cu after 50h CO₂RR (Figs. R4c and d).

This manuscript successfully synthesized dense vertical lamellate Cu nanostructures with excellent catalytic performance and prolonged stability. To explain the morphology stability of the DVL-Cu, COMSOL multiphysics simulations are carried out. Electrode current density and electrolyte current density of DVL-Cu and Cu₂O nanocubes during CO₂RR was acquired by COMSOL multiphysics simulations. Figs. R5a and c represent DVL-Cu while 5b and d represent Cu₂O nanocubes. Cu₂O nanocubes generate enhanced electrode current density with a nearly 5-fold increase. High-magnification images indicate that the electrode current density of regions far away from the substrate is higher than those of closer regions. This result could explain the

slight Cu agglomeration on the surface of nanoplates in DVL-Cu after CO₂RR. Meanwhile, the electrolyte current density (correlated to local electrostatic intensity) in the Cu₂O nanocubes system is higher than that in the DVL-Cu system, especially at the corner of nanocubes. Faster dissolution/redeposition process will occur in those regions with higher electrolyte current density, and increased local electrostatic intensity will lead to easier electro-migration of nanostructures. Simultaneously, the higher electrode current density of Cu₂O nanocubes could accelerate the agglomeration process. These simulation results explain why the Cu₂O nanocubes are easier to agglomerate during CO₂RR, which could be extrapolated to other easily-agglomerated nanostructures. Simultaneously, the moderate electrode current density coupled with the modest local electrostatic intensity ensures the structural stability of DVL-Cu during the CO₂RR. These microscopic-level analyses shed light on the structural stability of DVL-Cu.

Meanwhile, CuO nanoplates are grown in-situ on Cu foil or Cu-evaporated carbon paper. Better charge transfer and binder-free micro-circumstance would assist the catalytic process. The more robust combination with the substrate prevents the migration of Cu nanoplates, which helps avoid the agglomeration of DVL-Cu [Grosse, P. et al. *Angew. Chem. Int. Ed.* 57, 6192-6197 (2018)].

Secondly, the good stability of numerous delicate Cu/Cu₂O interfaces in KCl electrolyte is another key factor for its good performance. The choice of electrolytes is an important issue that can be easily overlooked. Significant differences exist between the catalytic performance obtained from the KHCO₃ system and non-buffering electrolytes, which could be discussed in detail in comment 3 (Fig. R6). More importantly, the high local pH generated by KCl electrolyte can help stabilize the delicate Cu₂O nanoparticles on the surface, where apparent agglomeration of Cu₂O nanoparticles in KHCO₃ is observed (See Figs. R8-10 and comment 3 for the details).

Cl⁻ specific adsorption further assists the suppression of hydrogen evolution in high negative bias during CO₂RR, resulting in higher C₂H₄ faradaic efficiency [Kotaro et al., *Electrochimica Acta*, 56, 2010, 381-386]. (See comment 3 for the details).

In short, the combination of delicate and stable nanostructures, three-dimensional self-supporting structure, and the utilization of non-buffering electrolyte are the most significant differences between this work and previous studies. These data were added in the revised manuscript (Supplementary Note 1) and the corresponding discussion is added in the main text.

Fig. R1 | **a,b**, TEM image of the CuO-NPs at different magnifications. **c**, AFM image of the CuO-NPs and the corresponding height profile from the dashed line. **d**, TEM image of the DVL-Cu.

Fig. R2 | **a,b**, SEM images of Cu₂O nanocubes at different magnifications. **c,d**, SEM images of Cu₂O nanocubes after 1h CO₂RR at different magnifications.

Fig. R3 | **a,b**, SEM images of thermal oxidation CuO at different magnifications. **c,d**, SEM

images of thermal oxidation CuO after 1h CO₂RR at different magnifications.

Fig. R4 | **a,b**, SEM images of CuO-NPs at different magnifications. **c,d**, SEM images of DVL-Cu after 50h CO₂RR at different magnifications.

Fig. R5 | **a,c**, Electrode current density and electrolyte current density distribution of DVL-Cu at 100 mA/cm². **c,d**, Electrode current density and electrolyte current density distribution of Cu₂O

nanocubes at 100 mA/cm^2 . The electrode current density and electrolyte current density (in **a** and **b**) correspond to the left and right legends, respectively. The electrode current density in **c** and **d** correspond to the left and right legends, respectively. The conical arrows represent the electrolyte current density vector.

2. Cu-based catalysts have been widely employed in CO_2 -to- C_2H_4 conversion so far, so I would be curious what are the intrinsic differences between this CuO system and previous reports [e.g., Dinh C T et al., Science, 2018, 360(6390): 783-787.] Indeed, the manuscript at present status cannot supply a solid explanation from the microscopic mechanism. In-depth research in this field should be necessary for publishing this work in Nature Communications.

We thank the reviewer for this question. As shown in Fig. R3, CuO systems commonly suffer from agglomeration issues. While some reports focused on improving the catalytic ability of three-phase boundary in the gas-diffusion electrode [e.g., Dinh C T et al., Science, 2018, 360(6390): 783-787.], our research is committed to building delicate nanostructures with stable Cu/Cu₂O interfaces for CO_2 -to- C_2H_4 catalysts. The reference is cited in the revised main text and the difference are discussed. As discussed in comments 1 and 3, we believe that the stable nanostructures (Fig. R4) and robust Cu(0)/Cu(I) interfaces (Fig. R10) are the intrinsic differences between this work and previous reports.

3. According to the Abstract in Line 38 and Line 87, the best performance was achieved in neutral KCl electrolyte, which may be more interesting than the performance data. As most of the reported highest performances were obtained in alkaline systems, which typically suffer from a severe carbonate side reaction. A well-established system in neutral electrolyte may be an important progress toward practical application of CO_2RR . But the authors only explained why KCl was chosen at present. The authors should provide a more detailed reason, as well as demonstrate a convincing mechanism on this phenomenon.

We thank the reviewer for pointing out this issue, which helps us understand the mechanism more deeply. To reveal the underlying mechanism of the electrolyte, we selected KHCO_3 (buffer electrolyte) and K_2SO_4 (non-buffer electrolyte without Cl^-) for analysis. Firstly, the catalytic performance of DVL-Cu in these electrolytes was measured. Total current density plots with

saturated CO₂ are shown in Fig. R6a. Current densities obtained from these electrolytes are very similar. The current density in K₂SO₄ is slightly higher than that in KHCO₃ and KCl. Besides, FEs of products may be more informative. The peak ethylene FE in KHCO₃ and K₂SO₄ is 66.4% (−0.7V) and 72.8% (−0.8V), respectively (Figs. R6b and c). Both ethylene FEs decline when the potential goes further negative, with increased hydrogen FEs. The methane production in K₂SO₄ is nearly negligible. In contrast, methane FE reaches 17.8% in KHCO₃ at −1.1V. By observing the overall product distribution, we can find that the ethylene production in KHCO₃ is not satisfied. The ethylene FEs exceed 60% only in the limited potential range and the methane production is not well suppressed at higher overpotentials. Ethylene FEs in K₂SO₄ at moderate overpotentials are similar to KCl, but hydrogen production dominates at higher overpotentials.

The performance difference can be explained by the following reasons. KHCO₃ is a buffering electrolyte and the local generated OH[−] during CO₂RR could be neutralized by HCO₃[−] leading to lower local pH than the non-buffering electrolyte. We conclude that the lower local pH in KHCO₃ results in inferior ethylene production because lower pH regions could be favorable to hydrogen evolution and methane production, both needing H⁺ [Hori et al., J. Chem. Soc., Faraday Trans. 1, 1989,85, 2309-2326]. Then, higher local pH generated in non-buffering electrolytes would suppress methane production in all potential ranges and hydrogen evolution at lower overpotentials. Comparing the catalytic performance in KCl and K₂SO₄, it could be concluded that the existence of Cl[−] could suppress hydrogen evolution even at high overpotentials, which is considered the result of the Cl[−] specific adsorption effect. The strongly adsorbed Cl[−] could facilitate the electron transfer from the electrode to CO₂ and suppress the adsorption of protons, leading to a higher hydrogen evolution overpotential [Kotaro et al., Electrochimica Acta, 56, 2010, 381-386].

Subsequently, long-term performances in different electrolytes were tested to explore the relationship between catalyst stability and electrolyte (Fig. R6d). Interestingly, the DVL-Cu delivered ~50h stable current density and ethylene FE in K₂SO₄ under −0.8V. In comparison, it only preserved stable performance for less than 10h in KHCO₃ under the same condition. SEM and TEM images of the post-CO₂RR sample in K₂SO₄ (Fig. R7) display equally morphology and Cu(I)/Cu(0) interfaces with that gained from the KCl sample, indicating that high local pH plays a critical role for the prolonged stability. Detailed post-electrolysis characterizations were

performed to unravel the underlying mechanism of stability difference for the sample obtained from KHCO_3 and KCl . As shown in Figs. R8c and d, a more severe Cu agglomeration on the top of nanoplates can be seen from the KHCO_3 sample. AC-TEM and EELS mapping were carried out to distinguish both samples' nanostructure and Cu(I) species distribution. Both samples are nanoplates composed of tiny nanoparticles with almost the same morphology (Figs. R9a, 10a). EELS mapping of the KHCO_3 sample (Figs. R9b-d) reveals that Cu(I) species agglomerate on the top of nanoplates while these species distribute uniformly in the KCl sample (Figs. R10d-f). AES depth profile analyses (Fig. R11) further verify the Cu(I) species distribution differences, where the Cu(I) content in KCl sample is higher along with depth compared to the KHCO_3 sample. Apparently, non-buffering electrolytes could stable the DVL-Cu catalyst by protecting its Cu(I)/Cu(0) interfaces during electrolysis. Since the K_{sp} of CuOH is relatively low (1.0×10^{-14}), high local pH could significantly slow down the dissolution of Cu(I) species. Hence, high local pH would suppress the dissolution/redeposition process of Cu(I) species in non-buffering electrolytes, which preserve the Cu(I)/Cu(0) interfaces during CO_2RR process.

Moreover, based on COMSOL simulation, the top of nanoplates possess a larger surface current density, which leads to higher local pH. It means the bottom region of nanoplates is easier to dissolve. The dissolved Cu(I) species would redeposit more likely on the top of nanoplates with higher local pH, which is consistent with SEM images of the KHCO_3 sample. These data were added in the revised manuscript (Note 1 in the Supplementary information) and the corresponding discussion is added in the main text and Supplementary Note 1.

Fig. R6 | **a**, FEs of the DVL-Cu in 0.5 M KCl. **b**, FEs of the DVL-Cu in 0.25 M K₂SO₄. **c**, FEs of the DVL-Cu in 0.5 M KHCO₃. **d**, LSV curves of DVL-Cu in 0.5 M KCl, 0.5 M KHCO₃ and 0.25 M K₂SO₄ during CO₂RR. **e**, Stability test of DVL-Cu in 0.5 M KCl, 0.5 M KHCO₃ and 0.25 M K₂SO₄ at -0.8V.

Fig. R7 | **a**, SEM image of the 50h post-electrolysis DVL-Cu in K₂SO₄. **b,c**, TEM images of the 50h post-electrolysis DVL-Cu in K₂SO₄ at different magnifications.

Fig. R8 | **a,b**, SEM image of the 50h post-electrolysis DVL-Cu in KCl at different magnifications. **c,d**, SEM images of the 20h post-electrolysis DVL-Cu in KHCO₃ at different magnifications.

Fig. R9 | **a**, STEM bright-field image of the 20h post-electrolysis DVL-Cu in KHCO₃. **b-d**, EELS

maps of Cu, Cu₂O and their overlay in the 20h post-electrolysis DVL-Cu in KHCO₃.

Fig. R10 | **a**, STEM bright-field image of the 50h post-electrolysis DVL-Cu in KCl. **b,c**, EDS maps of Cu, O element in the 50h post-electrolysis DVL-Cu in KCl. **d-f**, EELS maps of Cu, Cu₂O and their overlay in the 50h post-electrolysis DVL-Cu in KCl.

Fig. R11 | **a**, Cu LMM Auger spectra of the 20h post-electrolysis DVL-Cu in KHCO₃ with respect to different Ar⁺ etching depths. **b**, Cu LMM Auger spectra of the 50h post-electrolysis DVL-Cu in KCl with respect to different Ar⁺ etching depths.

4. As indicated in the article's title, the long-term stability is important for this work's novelty. However, the authors didn't give a critical analysis from the performance. For instance, why Cu(0)/Cu(I) interfaces, a thermodynamic factor according to authors' DFT calculations, could result in such a stable activity (i.e., dynamic phenomenon), which has not been reported in previous Cu/Cu₂O studies?

We thank the reviewer for pointing out the question. As revealed in comments 1-3, the long-term stability of DVL-Cu is derived from the following intrinsic factors: 1) Superior structural stability of DVL-Cu. The agglomeration of nanostructures is the main reason for poor stability in most cases. Therefore, superior structural stability is an essential precondition for long-term stability (as shown in Figs. R4b and c). 2) Correct electrolyte selection. As explained in comment 3 (Figs. R6-11), high local pH generated by non-buffering electrolytes effectively slows down the dissolution/redeposition process. Hence, Cu(I)/Cu(0) interfaces are well preserved in these electrolytes. These data were added in the revised manuscript (Fig. 6 in the main text, Supplementary Note 1) and the corresponding discussion is added in the Supplementary Note 1.

5. To follow up the previous question, what were the catalyst structures after the 55 hours of flow cell tests? Could these Cu(0)/Cu(I) interfaces be maintained after this long-term test at 150 mA cm⁻²? The authors should provide detailed post-electrolysis characterizations and give a clear explanation. Besides, EIS cannot supply much useful information in such a multistep reaction with unknown pathways.

We thank the reviewer very much for mentioning this important issue. We agree with the reviewer that EIS cannot supply much useful information in such a multistep reaction with unknown pathways. As requested by the reviewer, detailed post-electrolysis characterizations (SEM, TEM, AC-TEM, EELS mapping and AES depth profile) in KCl electrolyte coupled with failure analysis in KHCO₃ electrolyte were performed to unravel the underlying mechanism of stability. The nanoplates densely stack in order without any agglomeration after 50h electrolysis, only a slight increase of plates thickness is observed (Figs. R8a and b). AC-TEM images display that the nanoplates are composed of tiny nanoparticles, consistent with the 2h post-electrolysis sample. Elemental mapping shows the uniform distribution of Cu and O elements in the nanoplates (Figs. R10b and c). Valence state analyses exhibit numerous Cu(I)/Cu(0) interfaces without any

agglomeration (Figs. R10d-f). AES depth profiles (Fig. R11b) reveal that subsurface Cu(I) content is substantial, which is sharply contrasting with that in KHCO_3 . We argue that the moderate surface current density and local electrostatic intensity coupled with the stable self-standing nanostructures guarantee superior structural stability of DVL-Cu. In addition, high local pH generated by non-buffering electrolytes effectively slows down the dissolution/redeposition process, Cu(I)/Cu(0) interfaces are well preserved in these electrolytes. In brief, stable nanostructure and steady Cu(I)/Cu(0) interfaces result in the prolonged stability of DVL-Cu. These data were added in the revised manuscript (Note 1 in the Supplementary information) and the corresponding discussion is added in the main text.

6. In the Supporting information, the authors presented the 3D optical profiler simulated images and calculated results of CuO; and in Line 128, the authors stressed that “The Ra of CuO-NPs is 128 5.96 μm , dramatically larger than that of CuO (3.730 μm)”. Later in the text (Line 223), roughness factor (RF) was also calculated by double-layer capacitance method. So, what is the difference between ECSA and roughness factor? And what can these data be useful for the CO_2RR evaluation?

We thank the reviewer very much for mentioning this interesting issue. The Ra measured by 3D optical profiler is different from the roughness factor measured by the double-layer capacitance method. The former indicates physics roughness while the latter is equal to ECSA. ECSA is defined as the ratio of the reactive area of catalyst to that of pure copper. RF is obtained by dividing the double-layer capacitance (equal to the reactive area) of the catalyst by the double-layer capacitance of ideal smooth copper, which is a standard method to calculate ECSA [Sachin et al., *Advanced Powder Technology* 29 (2018) 3520–3526]. Therefore, RF is the result of calculated ECSA. For CO_2RR catalysts, delicate nanostructures could increase the number of electrochemically active sites, increasing ECSA value and improving reaction activity. Hence, ECSA can be used as a reference standard for the reaction activity of catalysts. On the other hand, physics roughness (Ra) can be used as an indicator of ECSA in most cases, and usually a higher physics roughness (Ra) means increased ECSA. Thus, we further used the Ra to illustrate the exquisite nanostructures of the designed catalysts.

Reviewer #2 (Remarks to the Author):

This paper presents a combined experimental and DFT study of CO₂ electrochemical conversion to C₂ products on CuO/Cu catalysts. The authors need to address the following points before it is considered as a potential Nat. Commun publication:

(i) The authors claimed that C₂H₄ is the main product. However, Figure 4e clearly indicates that ethanol is the main product. The same figure also shows that formate is one of the main products. There is no C₂H₄ in Figure 4c.

We thank the reviewer for pointing out the question. The authors would like to display ratios of liquid products in Figure 4e. However, this form was inappropriate due to the lack of Faradic efficiencies and the misleading expression. We replaced the “ratios” in the Y-axis of Figure 4e to “FEs”, which clearly shows the precise liquid FEs of catalysts at different potentials and effectively avoids misunderstanding (Fig. R12). These data were updated in the revised manuscript (Fig. 4e in the main text) and the corresponding discussion is added in the main text.

Fig. R12 | Liquid products FEs of DVL-Cu and R-CuO-NPs.

(ii) As the authors mentioned HER is always a competing reaction. The HER free energy diagram is not included and discussed in the manuscript. The authors should perform DFT to get HER energetics and should be compared with the CO₂RR free energy diagrams to show that HER is less favorable and the catalysts selectively promote the CO₂RR.

We thank the reviewer for pointing out this question. We have calculated HER energetics of three

catalytic sites and compared them with CO₂RR free energy diagrams (Fig. R13). DFT results present that the HER barriers of three catalytic sites (0.82 eV on Cu(110) slab, 1.44 eV on Cu/Cu₂O interface and 0.72 eV on Cu₂O(110) slab) are higher than the ethylene rds (rate-determined step) free energy barriers of Cu/Cu₂O site (0.60 eV). Therefore, it can be proved that CO₂RR is more favorable than HER in the DVL-Cu system. Also, the HER barrier of Cu/Cu₂O interface is the highest, consistent with the experimental results that H₂ FE of DVL-Cu is the lowest. These data were updated in the revised manuscript (Fig. 54 in the Supplementary information) and the corresponding discussion is added in the main text.

Fig. R13 | A reaction energy diagram for HER on Cu(110) slab, Cu/Cu₂O interface and Cu₂O(110) slab.

(iii) As shown in Figure 4c, the authors should also include and discuss the formate pathway and compared with the HOCO pathway which is claimed to produce C₂ products.

We thank the reviewer very much for mentioning this important issue. We have calculated different paths of the formate formation process on three catalytic sites and compared them with free energy barriers of C₂ pathway (Fig. R14), such as *CO₂+H→*HCOO (ΔG = 0.80 eV on the Cu/Cu₂O interface), *CO₂+H→*COOH (ΔG = 0.50 eV on the Cu/Cu₂O interface), *COOH+H→*HCOOH (ΔG = 1.57 eV on the Cu/Cu₂O interface) and *COOH+H→*CO (ΔG = -0.32 eV on the Cu/Cu₂O interface). By comparing their free energy barriers, it indicates that the energy barriers of formate formation (through *HCOO or *COOH intermediate) are higher than

that of C2 product pathway. Finally, it is found that the pathway on the Cu/Cu₂O interface that leads to the *OCCOH formation (C2 product intermediate) is the optimal one. Therefore, it can be proved that C2 products are more favorable than formate in the DVL-Cu system. These data were updated in the revised manuscript (Fig. 53 in the Supplementary information) and the corresponding discussion is added in the main text.

Fig. R14 | A reaction energy diagram for CO₂RR on Cu(110) slab, Cu/Cu₂O interface and Cu₂O(110) slab. Intermediates corresponding to lighter colors are marked in red font.

(iv) Since ethanol is one of the main C₂ products, it would be helpful to have free energy diagram that leads to the formation of C₂H₅OH and be compared with the C₂H₄ pathway.

We thank the reviewer very much for mentioning this important issue. We have calculated the catalytic path of ethanol (*OHC₂H₂→*OHC₂H₃→*OHC₂H₄→*+OHC₂H₅) on three catalytic sites (Fig. R15). After comparison, the first divergent step (*OHC₂H₂→*OHC₂H₃, ΔG = 2.32 eV on Cu/Cu₂O interface) of ethanol is unfavorable to proceed compared with ethylene pathway (*OHC₂H₂→*O+C₂H₄, ΔG = -0.52 eV on Cu/Cu₂O interface) on three catalytic sites. Therefore, DFT results match well with the low FE of ethanol (< 7%) in the DVL-Cu system. These data were updated in the revised manuscript (Fig. 55 in the Supplementary information) and the corresponding discussion is added in the main text.

Fig. R15 | A reaction energy diagram for CO₂RR to ethylene and ethanol on Cu(110) slab, Cu/Cu₂O interface and Cu₂O(110) slab. Intermediates corresponding to lighter colors are marked in red font.

(v) The authors assumed that CH₄ forms from COH intermediate. However there are several other channel that lead to the formation of CH₄. For example: $\text{CO}_2 \rightleftharpoons \text{HCOO} \rightleftharpoons \text{HCOOH} \rightleftharpoons \text{H}_2\text{COOH} \rightleftharpoons \text{H}_2\text{CO} \rightleftharpoons \text{H}_3\text{CO} \rightleftharpoons \text{CH}_3 \rightleftharpoons \text{CH}_4$. There is no rational presented why other channels are not included in the discussion.

We thank the reviewer for pointing out the question. We do not explain this clearly in the first version. CO₂RR is a continuous proton-coupled electron transfer (PCET) process. In a single hydrogenation step, there are several reaction sites, which will lead to different products ultimately. Free energy barriers of different hydrogenation steps that lead to methane and C₂ products were calculated on three catalytic sites. As shown in Fig. R14, the first PCET step that form *COOH intermediate requires lower free energy than that form *HCOO intermediate, which indicates the PCET process tends to occur through *COOH intermediate on three catalytic sites. Based on the optimal intermediate of the previous step, the free energy barriers of *CO and *HCOOH intermediate generated from *COOH have been calculated to distinguish the reaction trend. Combining the DFT results, the CO₂RR tends to process through: $\text{CO}_2 \rightleftharpoons \text{*COOH} \rightleftharpoons \text{*CO} \rightleftharpoons \text{*CO+CO} \rightleftharpoons \text{*CO+COH} \rightleftharpoons \text{*OCCOH}$ (C₂ pathway) on three

catalytic sites. Meanwhile, the most possible CH₄ pathway (if proceed) is CO₂==>*COOH==>*CO==>*COH. However, insurmountable energy barriers are required for the hydrogenation of *CO to form *COH (*CO+H→*COH, reaction step of CH₄ formation pathway) on three catalytic sites, which leads the CO₂RR to C₂ products instead of CH₄. These data were updated in the revised manuscript (Fig. 53 in the Supplementary information) and the corresponding discussion is added in the main text.

(vi) The selection of DFT model for the interface needs to be justified. Did authors test other models (smaller/larger nanoparticle)? What is the size of Cu₂O nanoparticle in experiment? Does the DFT model correspond to experiments?

We thank the reviewer very much for pointing out this important issue. We have validated the computational model when constructing the model. Firstly, we calculated the adsorption area ratio of Cu₂O nanoparticles (~48%), Cu/Cu₂O (~6.18:1) ratio and the size of Cu₂O nanoparticles (4-12 nm) from the experimental results. Different expansion ratios of Cu(110) surface, including 3*3, 4*4, 5*5, 3*6, were simulated to verify their stability and structural property. Subsequently, it is found that to achieve the surface adsorption ratio of Cu₂O nanoparticles (50%) and Cu/Cu₂O ratio (6:1), the expansion ratio of 3*6 (16.58 Å*8.31 Å*25.84 Å) is the most appropriate. Scaling down the model equally would lead to structural instability of Cu/Cu₂O interfaces while scaling up the model would exceed the state-of-the-art computational capability. In fact, the 3*6 model possesses all the structural characteristics and identical interface properties with the DVL-Cu system. Therefore, the DFT model for the interface is consistent with experimental results and could demonstrate catalytic performance of the DVL-Cu catalyst.

Reviewer #3 (Remarks to the Author):

Wei Liu et al., presented a facile synthesis method to prepare Cu-based electrocatalyst for the CO₂RR. They have shown that their catalyst, called dense vertical lamellate Cu nanoparticles (DVL-Cu), under the reduction applied potential maintains the Cu₂O/Cu interface to keep ethylene Faradaic efficiency (FE) at high value for a long time, while with a considerably improved total energy efficiency (EE). The attained FE for ethylene is 84.5%, and stable for 55 hours, which is a

significant improvement compared to the previous reports so far. A thorough set of characterizations is performed to shed light on the relation between catalyst composition/structure and its superior activity. Overall, this reviewer is confident to recommend the manuscript to be published in the journal Nature Communications, after considering the following suggestions.

1. In page 12, line 252, it is recommended to add a SEM image of DVL-Cu after 2h CO₂RR (and through comparison) showing that the agglomeration of Cu nanoparticles is the main reason for the instability.

We thank the reviewer very much for pointing out this important issue. We compared the SEM images of DVL-Cu and R-CuO-NPs taken after 2h CO₂RR. As shown in Figures R16c and d, the agglomeration of the nanoplates to nanoparticles is observed after 2h CO₂RR of R-CuO-NPs. In contrast, the SEM images of DVL-Cu (Figures R16a and b) indicate that the vertically arranged and densely stacked laminated nanostructures are retained after 2h electrochemical reduction, which shows the agglomeration of Cu nanoparticles is the main reason for the instability. These data were added in the revised manuscript (Fig. 22 in the Supplementary information) and the corresponding discussion is added in the main text.

Fig. R16 | **a,b**, SEM image of the 2h post-electrolysis DVL-Cu at different magnifications. **c,d**, SEM image of the 2h post-electrolysis R-CuO-NPs at different magnifications.

2. The author emphasized the capability of this method for industrialization, yet using KCl as the

electrolyte is quite hazardous, and it might be a big challenge for industrialization.

We thank the reviewer for pointing out the question. We agree with the reviewer that chloride ion (Cl^-) could discharge at the anode in KCl electrolyte, harmful to the device and environment. We did observe that the Cl^- in catholyte would be transported through the membrane to anolyte when anion exchange membrane (AEM) was used in the flow cell system. Fortunately, although KCl as catholyte was used, the anolyte choice of H-cell system was arbitrary without difference in catalytic performance. Meanwhile, the Nafion-117 membrane can be used as the membrane, which can completely block Cl^- transmission. As shown in Figure R17, the Nafion membrane system display nearly the same FEs of ethylene and full-cell EEs at different current densities with the AEM system (KHCO_3 was chosen as the anolyte). Therefore, the catalytic performance of DVL-Cu was irrelevant with the anolyte and ion exchange membrane. Those anolytes with no Cl^- and membranes that could block the transportation of Cl^- would be suitable for industrialization in the future. These data were added in the revised manuscript (Fig. 34 in the Supplementary information) and the corresponding discussion is added in the Supplementary information.

Fig. R17 | Ethylene FE and EE of DVL-Cu in flow cell using Nafion membrane.

3. In Fig. 4e, it is recommended to plot FE rather than ratios, albeit the FEs are quite small. Otherwise, this reviewer does not find the figure informative. Also, please use the same potential range for Fig. 4A and B.

We thank the reviewer for pointing out this. The liquid FEs of DVL-Cu and R-CuO-NPs at

different potentials are plotted in Figure R18. Meanwhile, the same potential ranges from -0.6 V to -1.1 V were chosen in Figs. 4A and B to compare their catalytic performance. Now, this figure is more informative (Fig. R19). These data were updated in the revised manuscript (Figs. 4a and 4e in the main text) and the corresponding discussion is added in the main text.

Fig. R18 | FEs of liquid products of DVL-Cu and R-CuO-NPs.

Fig. R19 | FEs of DVL-Cu.

4. In line 230 and 231, the author mentioned the generation of H_2 is severely suppressed. 21% FE is not considered a ‘severely’ suppression. Also, total FEs are not reported. From the current results it seems the total FE goes above 100%. In addition, the rational provided in line 232 for the H_2 suppression is not satisfying. This reviewer suggests testing both catalysts for the HER and compare the results. It is possible that the DVL-Cu intrinsically is less active for the HER.

We thank the reviewer very much to mention this important issue. HER tests were performed on

both catalysts, and results were plotted with LSV curves obtained during CO₂RR. As shown in Fig. R20, the current density of DVL-Cu during CO₂RR is much higher than that in the HER condition, indicating intrinsic higher CO₂RR activity than HER. Current densities of R-CuO-NPs have a similar trend, but the enhancement is much smaller than that of DVL-Cu. From Fig. R20 we can conclude that the two catalysts have similar intrinsic HER activity, while the DVL-Cu has much higher intrinsic CO₂RR activity than R-CuO-NPs. Thus, the high ethylene FE of DVL-Cu is originated from the higher intrinsic CO₂RR activity rather than the suppression of HER. We changed the description of “severely suppression of H₂” to “the FE_{H₂} is only 13.6% at -0.9 V of DVL-Cu”.

In addition, we carefully checked FEs and found that only the total FEs at -0.9V of DVL-Cu was slightly beyond 100%. This phenomenon violates scientific principles but was common in the CO₂RR research field (for example, in *Science* 372, 1074 2021). The errors usually arise from the quantification of gaseous and liquid products. We retested the potential whose total FE went above 100% and list the new total FEs in Table 1. These data were updated in the revised manuscript (Fig. 4a in the main text, Fig. 18 and Table 1 in the Supplementary information) and the corresponding discussion is added in the main text.

Fig. R20 | LSV curves of DVL-Cu and R-CuO-NPs in Ar and CO₂ saturated KCl electrolyte.

Table R1 | FEs of DVL-Cu at different potentials in H-cell.

Potential (V)	FE (%)									
	C ₂ H ₄	H ₂	CO	CH ₄	formate	ethonal	acetate	Pr-OH	C ₂ H ₆	Total
-0.6	43.8	39.7	10.8	0.0	1.6	1.5	0.1	0.5	0.0	97.9
-0.7	63.9	28.1	1.4	0.7	0.6	3.0	0.1	0.4	0.0	98.1
-0.8	69.8	20.4	2.6	0.3	0.9	2.8	0.1	0.8	0.8	98.5
-0.9	74.9	13.6	2.9	0.8	1.0	3.3	0.1	0.8	1.4	98.9
-1.0	66.3	17.2	2.6	4.8	0.7	4.8	0.1	0.8	1.9	99.2
-1.1	60.8	22.3	1.5	5.9	1.4	5.5	0.0	0.4	0.5	98.3

5. In lines 248-252: why two catalysts are tested at two different potentials for the stability test? They must be compared under similar conditions, otherwise, drawing a conclusion would not be possible. Either test either DVL-Cu at -1.0V or R-CuO-NPs at -0.8V.

We thank the reviewer for pointing out the questions. It is our fault for testing the two catalysts at different potentials for the stability test. We tested R-CuO-NPs at -0.8V and depicted the results in Fig. R21. As shown in Fig. R21, the DVL-Cu delivers a steady i-t curve and high C₂H₄ selectivity for 50h at -0.8V. In contrast, the R-CuO-NPs only retains stable FEC₂H₄ for less than 7h. These data were updated in the revised manuscript (Fig. 21 in the Supplementary information) and the corresponding discussion is added in the main text.

Fig. R21 | Stability test of DVL-Cu and R-CuO-NPs at constantly applied potentials.

6. Lines 253-261: although Cu₂O was shown in the HRTEM for DVL-Cu, it is recommended adding GI-XRD profiles after 1h CO₂RR, like what has been done for the R-CuO-NPs.

As requested by the reviewer, GI-XRD profiles after 1h and 2h CO₂RR were provided in Fig. R22, and Fig. 3a in the main text, respectively. As shown in these images, the Cu₂O characteristic peak remains after 1h or 2h CO₂RR, which is absent in the R-CuO-NPs profiles after 1h CO₂RR. Meanwhile, the peak intensity of R-CuO-NPs is higher than that of DVL-Cu, indicating that it has a larger crystal size. This could correspond to Cu nanoparticles' agglomeration of R-CuO-NPs, as shown in Fig. R16. These data were updated in the revised manuscript (Fig. 24 in the Supplementary information) and the corresponding discussion is added in the Supplementary information.

Fig. R22 | GI-XRD profile of the R-CuO-NPs and DVL-Cu taken after 1 h CO₂RR.

7. In the table 1, the partial current density of C₂H₄ was mentioned 174.4 mA/cm² which does not seem to be super accurate. At -0.81V the partial current density was mentioned 92.5 mA/cm² (line

289).

We thank the reviewer for this question. The current density of C₂H₄ does have some mistakes because it reaches 174.4 mA/cm² at -1.01V instead of -0.81V. The author's original purpose was to show the maximum current density of C₂H₄ but these means of expression were easy to be misunderstood. Now we update Table R2, in which the potentials, FEs and current densities correspond to each other. These data were updated in the revised manuscript (Table 1 in the main text) and the corresponding discussion is added in the main text.

Catalysts	Electrolyte	Potential (V vs. RHE)	FE _{C₂H₄} (%)	j _{C₂H₄} (mA cm ⁻²)	Maximum EE _{C₂H₄} (%)*	Reference
DVL-Cu	0.5 M KCl	-0.81	84.5	92.5	28	This work
		-1.01	67.1	174.4		
OBC	0.5 M KHCO ₃	-1.00	45	44.7	/	20
O ₂ -plasma-treated Cu	0.1 M KHCO ₃	-0.9	60	7.2	/	21
Cu 3D CTPI	/	/	69	304	22	51
Cu(B)-2	0.1 M KCl	-1.1	52	36.4	/	14
Cu-12	1 M KHCO ₃	-0.83	72	230	20	52

*The maximum EE_{C₂H₄} was not achieved in the potentials listed in the table.

8. Line 299, the author claimed this is the highest value achieved in the ‘‘neutral’’ catholyte. This requires further evidence, e.g., measuring pH, during and (especially after the long stability experiment). A plot of pH vs. time will be very informative.

We thank the reviewer very much to mention this important issue, which makes our work more solid. As requested by the reviewer, pH value of the catholyte was measured during long-term stability experiment. In the H-cell system (Fig. R23), pH value raised rapidly from 3.90 to 6.52 in the initial 1h. Then pH value retained around 6.7 for the following test, a neutral catholyte environment. Meanwhile, the FE of C₂H₄ remained stable in this pH range, indicating good catalytic performance could be achieved in the neutral catholyte.

Moreover, during the long-term stability experiment of the flow cell system, the catholyte was replaced every 6h in the previous test. To monitor the variation of pH value of catholyte during the test, the catholyte remained unchanged for 24h. As shown in Fig. R24, pH value raised from 6.95 to 8.23 in the first 1h, and reached 9.50 after 5h electrolysis. The pH value fluctuated around 9.8 for the rest of time. Despite pH changing over time, the FE of C₂H₄ remained stable around 80%, demonstrating that DVL-Cu delivers similar catalytic performance in the pH range from 6.5 to 9.8. In industrial production, the pH of the solution can be easily adjusted near neutral, which could minimize corrosion to equipment without affecting the yield. These data were updated in the revised manuscript (Figs. 19 and 42 in the Supplementary information) and the corresponding discussion is added in the Supplementary information.

Fig. R23 | pH and ethylene FE during the long-term test in H-cell.

Fig. R24 | pH and ethylene FE during the long-term test in flow cell.

9. The manuscript needs another round of proofreading as there are several typos and sentences with grammatical errors which make it difficult to understand. Few examples are mentioned below:

- Line 32: “an facile”
- Fig. 1c: CuO growth and CuO-nucleation graph caption should be revised.
- Supplementary Fig. 10 caption: R-CuO instead of A-CuO. This is repeated many times.
- Line 93: “catholic” reduction
- Supplementary Fig. 20 caption is incorrect.

We thank the reviewer very much. The above mistakes have been completely corrected now.

- Line 150: the authors discuss the decrease in Cu (110) peak intensity but no peak is assigned to this facet in Fig. 2d.

Copper belongs to the face-centered cubic system. Cu (110) peak would be eliminated in X-ray diffraction. Cu (220) facet is the equivalent crystal plane of Cu (110), which has the same relationship as Cu (200) and Cu (100) in Fig. 2d. Therefore, we could judge the peak intensity variation trend of Cu (110) from that of Cu (220) in Fig. 2d.

- Fig. 2F: don't we expect to observe an intensified Cu₂O peak at -1V compared to that at -0.6V after 1800 s?

Cu₂O would be gradually reduced to Cu⁰ under negative potentials according to electrochemical principles. Either extending the reducing time or improving the negative bias will lead to the

decrease of the Cu₂O ratio in the catalyst, which could also be seen in previous studies like **Nature Catalysis** volume 1, pages 103–110 (2018). Therefore, it is reasonable that weaker Cu₂O was seen at $-1V$ compared to $-0.6V$.

REVIEWERS' COMMENTS

Reviewer #1 (Remarks to the Author):

This revised manuscript at current stage shows good improvement on the study of CuO-based CO₂RR and reveals the atomic-scale mechanism (i.e., Cl⁻ anion-induced stable Cu(0)/Cu(I) interface as well as higher local pH) for efficient CO₂-to-C₂H₄ conversion. As a result, from the perspective of internal mechanism, research novelty, and methodology, the revised manuscript can be accepted in Nature communication with the following minor revision.

Suggestion: Based on the response by the authors, "physics roughness (Ra) can be used as an indicator of ECSA in most cases, and usually a higher physics roughness (Ra) means increased ECSA", we would conclude that physics roughness (Ra) plays almost the same role with ECSA. According to Occam's Razor, the concept of Ra is not necessary to be proposed in this work when ECSA can be obtained. We'd suggest authors to delete the Ra-related description and rewrite this part to make this work more concise. Otherwise, readers outside this field may be confused.

Reviewer #2 (Remarks to the Author):

The authors have addressed most of the questions. However, the line of argument presented for the DFT model construction of Cu₂O/Cu is not convincing. The binding strength of intermediates depend on the side of the Cu₂O cluster on Cu surface. The authors should test few different clusters of different size and calculate the CO₂RR free energy diagrams to show that the DFT conclusions remain same and agree with the experimental trend.

Reviewer #3 (Remarks to the Author):

none

Reply to reviewers

Reviewer #1 (Remarks to the Author):

This revised manuscript at current stage shows good improvement on the study of CuO-based CO₂RR and reveals the atomic-scale mechanism (i.e., Cl⁻ anion-induced stable Cu(0)/Cu(I) interface as well as higher local pH) for efficient CO₂-to-C₂H₄ conversion. As a result, from the perspective of internal mechanism, research novelty, and methodology, the revised manuscript can be accepted in Nature communication with the following minor revision.

Suggestion: Based on the response by the authors, “physics roughness (Ra) can be used as an indicator of ECSA in most cases, and usually a higher physics roughness (Ra) means increased ECSA”, we would conclude that physics roughness (Ra) plays almost the same role with ECSA. According to Occam's Razor, the concept of Ra is not necessary to be proposed in this work when ECSA can be obtained. We'd suggest authors to delete the Ra-related description and rewrite this part to make this work more concise. Otherwise, readers outside this field may be confused.

A: We thank the reviewer very much for providing this suggesting. According to Occam's Razor, the concept of Ra mentioned in the manuscript would confused readers while the ECSA appears simultaneously. Hence, we delete the Ra-related description and data in the revised manuscript and Supplementary information and rewrite this part to emphasize the concept of ECSA.

Reviewer #2 (Remarks to the Author):

The authors have addressed most of the questions. However, the line of argument presented for the DFT model construction of Cu₂O/Cu is not convincing. The binding strength of intermediates depend on the side of the Cu₂O cluster on Cu surface. The authors should test few different clusters of different size and calculate the CO₂RR free energy diagrams to show that the DFT conclusions remain same and agree with the experimental trend.

A: We thank the reviewer for pointing out this question. We understand the reviewer's concerns that the binding strength of intermediates depend on the side of the Cu₂O cluster on Cu surface, especially its size. At the request of the reviewer, the composite Cu/Cu₂O structures with four different sized Cu₂O clusters, including 6 Å*8 Å with 5 Cu₂O, 10 Å*8 Å with 8 Cu₂O (the model used in the manuscript), 14 Å*8 Å with 12 Cu₂O, and 16 Å*8 Å with 16 Cu₂O (as shown in Fig. R1b-e) was constructed. Refer to our previous DFT calculation results, the increased adsorption energy of the post-dimerization intermediate (*OCCOH) at Cu/Cu₂O interface reduce the energy barrier of C-C coupling step (the most important step of C₂ production), which is the essential origin of the superior catalytic performance of DVL-Cu. Meanwhile, the *CO+*CO→*CO+*COH is the rate-determining step (RDS) of ethylene production at the Cu/Cu₂O interface and Cu(110) slab. Hence, we calculated the free energy of the following reaction steps (as shown in Fig. R1a): *CO→*CO+*CO→*CO+*COH→*OCCOH. DFT calculation results reveal that the free energies of C-C coupling steps are exergonic on all

Cu/Cu₂O models of different cluster sizes, indicating that the cluster sizes have no significant impact on the adsorption capacity of the post-dimerization intermediate at the Cu/Cu₂O interfaces. Moreover, the energy barriers of *CO+*CO→*CO+*COH are very similar for these four different models (0.59-0.66 eV), lower than that on the Cu(110) slab (1.15 eV) and Cu₂O(110) slab (0.75 eV) and the RDS of other competing by-products, indicating that the size of Cu₂O cluster will not change the conclusion of this work. Furthermore, as the C₂H₄ desorption is the RDS of ethylene production on Cu₂O(110) and is exergonic only on Cu/Cu₂O interfaces (Cu₂O size: 10 Å*8 Å) according to the previous calculation results, the reaction free energies of C₂H₄ desorption were also calculated. As shown in Table R1, the reaction free energies are exergonic on Cu/Cu₂O interfaces of different Cu₂O cluster sizes, indicating that the Cu/Cu₂O interfaces of different Cu₂O cluster sizes possess a fast C₂H₄ production capacity equally. These results suggest that the DFT conclusions remain same and agree with the experimental trend when altering the size of adsorbed Cu₂O clusters. We sincerely hope that these additional experiments could address your concerns. These data were added in the revised manuscript (line381-382, Supplementary Fig. 47 and Supplementary Table 1) and the corresponding discussion was added in the Supplementary Note 3.

Fig. R1 a A reaction energy diagram for *CO→*OCCOH on the Cu/Cu₂O interface of different Cu₂O cluster sizes. b-e Geometry of Cu(110)/Cu₂O(110) configurations of different Cu₂O cluster sizes for DFT calculations.

Table R1. Reaction free energy of C₂H₄ desorption on different reaction sites.

Reaction sites	Reaction free energy of C ₂ H ₄ desorption
Cu(110)	0.46 eV
Cu ₂ O(110)	1.19 eV
Cu/Cu ₂ O (6 Å*8 Å)	-0.44 eV
Cu/Cu ₂ O (10 Å*8 Å)	-0.52 eV
Cu/Cu ₂ O (14 Å*8 Å)	-0.49 eV
Cu/Cu ₂ O (16 Å*8 Å)	-0.57 eV

Reviewer #3 (Remarks to the Author):

None

A: We thank the reviewer for his/her revised view of our manuscript and as no comments for improvement were made we did not change the manuscript.